# Far-East Asian *Toxoplasma* isolates share ancestry with North and South/Central American recombinant lineages

Fumiaki Ihara [1,2,3], Hisako Kyan[4], Yasuhiro Takashima[5,6], Fumiko Ono[7], Kei Hayashi [8], Tomohide Matsuo[9], Makoto Igarashi[10], Yoshifumi Nishikawa[10], Kenji Hikosaka[11], Hirokazu Sakamoto [11], Shota Nakamura[12], Daisuke Motooka[12], Kiyoshi Yamauchi[13], Madoka Ichikawa-Seki[14], Shinya Fukumoto[10], Motoki Sasaki[15], Hiromi Ikadai[16], Kodai Kusakisako[16], Yuma Ohari[17], Ayako Yoshida[18,19], Miwa Sasai [1,2,3], Michael E. Grigg[20] & Masahiro Yamamoto [1,2,3] ✉

*Toxoplasma gondii* is a global protozoan pathogen. Clonal lineages predominate in Europe, North America, Africa, and China, whereas highly recombinant parasites are endemic in South/Central America. Far East Asian *T. gondii* isolates are not included in current global population genetic structure analyses at WGS resolution. Here we report a genome-wide population study that compared eight Japanese and two Chinese isolates against representative worldwide *T. gondii* genomes using POPSICLE, a novel population structure analyzing software. Also included were 7 genomes resurrected from nonviable isolates by target enrichment sequencing. Visualization of the genome structure by POPSICLE shows a mixture of Chinese haplogroup (HG) 13 haploblocks introgressed within the genomes of Japanese HG2 and North American HG12. Furthermore, two ancestral lineages were identified in the Japanese strains; one lineage shares a common ancestor with HG11 found in both Japanese strains and North American HG12. The other ancestral lineage, found in *T. gondii* isolates from a small island in Japan, is admixed with genetically diversified South/Central American strains. Taken together, this study suggests multiple ancestral links between Far East Asian and American *T. gondii* strains and provides insight into the transmission history of this cosmopolitan organism.

*Toxoplasma gondii* is a worldwide parasite infecting an estimated 30% of the global population[1]. Previous studies have shown the global diversity of *T. gondii* in Europe, the Americas, and Africa. Archetypal lineages I, II, and III are predominant in Europe[2], and exist as recombinant lineages that expanded clonally and are distributed globally[3]. The *T. gondii* population genomic structure is comprised of 6 major clades that are subdivided into 16 major haplogroups that cluster together in a distance tree[4]. The three archetypal lineages now belong to Haplogroups (HGs) 1, 2, and 3. Geographic segregation of the other 13 haplogroups is common, with for example type X belongs to HG12 mostly isolated from North America, whereas HG11 is largely restricted to wild felids[5–7]. The genetic diversity of *T. gondii* in South/Central America is quite high and differs markedly from Europe and North America[8]. Although common clonal lineages (Africa 1 in Brazil, and type II in Chile) exist, highly genetically diverse recombinant strains are expanding[8]. Archetypal lineages II and III are also the major lineages in

Africa, Africa 1 circulates widely in the West and Central areas, belonging to HG6[8]. Africa 3 that might have emerged after two or more recombination events between Africa 1, type II, and type III strains, belonging to HG14, are found in Gabon[9,10]. Another genotype Africa 4, referred to as, Restriction Fragment Length Polymorphism (RFLP) #20 has been found in Egypt, Tunisia, Senegal, and Benin[11,12]. Strains with this genotype were also isolated in Sri Lanka, Arab Peninsula, and China[13]. In recent years, genomes of numerous *T. gondii* isolates have been sequenced. A haplotype map divided the entire genome into 157 blocks based on pairwise SNP comparisons[14]. Recently derived recombinant strains significantly contribute to the current population of isolated *T. gondii*, aside from those strains that originate from distant locales and wild felines[14]. Therefore, it is essential to identify how recombination has impacted the population genomics of currently circulating strains to independently estimate the evolutionary history of each haploblock[14]. A study with 62 isolates with higher sequencing depth data was reported in 2016[4]. Most recently, the largest population analysis with the whole genomes of 156 isolates including many African and Caribbean isolates was conducted[15].

Most molecular epidemiologic studies of *T. gondii* in Asian countries have been conducted in China and have been of limited scope. A previous meta-analysis showed that 66.5% of a total of 278 *T. gondii* isolates analyzed across China belonged to HG13[16]. HG1 was the second most prevalent HG (20.6%), followed by HG2 (7.2%). In aggregate, these three HGs accounted for almost 95% of the diversity found in China. However, genotyping was largely limited to only a few markers, so it remains unclear which of the haplogroups each isolate truly belongs too. Nevertheless, many studies suggest that strains with HG1 alleles predominate. In Korea, the KI-1 is known to be HG1[17]. In Japan, two strains identified as HG2, namely TgCatJpTy1/k3 and Fukaya, were isolated several decades ago. In addition, 24 strains were successively isolated in Okinawa, a southern part of Japan, (Supplementary Fig. S1), between 2008–2013[18,19]. An atypical strain TgCatJpGi1/TaJ which was isolated in central Honshu, the main island of Japan (Supplementary Fig. S1), showed moderate virulence in mice despite it being described as HG3, a mouse avirulent lineage[20]. Interestingly, two strains (TgCatJpObi1 and TgMonkeyJp1) which were isolated in Hokkaido, the north island of Japan (Supplementary Fig. S1), possessed the same RFLP genotype as RFLP#4 mainly found in domestic animals in North America[21,22]. A recent study investigating the genotypes of 15 isolates from Okinawa based on the alleles found for three protein-coding genes plus three intron sequences, suggested that the seven strains clustered closely with HG2, while one strain was closely related to HG3[23]. The remaining seven isolates formed their own cluster flanked by CAST (HG7), isolated from a US AIDS patient. Thus, these limited genotyping studies suggested that the *T. gondii* population from Far East Asia may be distinct and possess a complex genetic diversity with potential links to *T. gondii* genomes from other regions of the globe. 'Far East Asia' often encompasses countries in Southeast Asia. However, in our study, the term 'Far East Asia' specifically refers to Japan, China, South Korea, Mongolia, and Taiwan, and excludes Southeast Asian countries such as Indonesia, Malaysia, and the Philippines. This study investigated the population genomic structure of Far East Asian *T. gondii* to investigate how they relate to *T. gondii* strains in other regions and whether they consist of known or unique ancestral lineages. To facilitate these analyses, we utilized a software suite known as POPSICLE to comprehensively visualize the extent of recombination between lineages. POPSICLE resolves the genome into small blocks, identifies the number of ancestral lineages in each block, and agnostically paints each ancestry with a distinct color to resolve haploblocks with shared ancestry. Using POPSICLE, recombination between subspecies can be visualized, and produce models that resolve the extent to which intra- and inter-specific recombination is occurring between species complexes, as was recently demonstrated for the Ascaris species complex[24]. In this study, we utilized POPSICLE to investigate

genome-wide SNP diversity in Japanese *T. gondii* in the context of all major *T. gondii* HGs plus Africa 4 (#20), including isolates from China. Using this approach, we identified a recombinant ancestry among circulating strains in Far East Asia, and specifically demonstrated the existence of these same haploblocks circulating among strains isolated from North and South/Central America. Collectively, these findings support a population structure that is fluid, with hallmarks of recombination followed by geographic segregation that are likely selected for their transmission, host range, and pathogenesis at a local scale.

## Results

### Genome-wide phylogenetic analysis identified frequent admixture patterns in *Toxoplasma* population in Japan

To investigate the population structure of *T. gondii* in Japan at a genome-wide resolution, we constructed a large dataset comprised of 57 strains in total. This dataset included 2 or 3 representative strains from the 16 haplogroups plus 3 Africa 4 (#20) strains along with 8 Japanese strains, including 6 newly sequenced strains (Supplementary Fig. S1 shows the regions where they were isolated). Additionally, we resurrected the genomes of seven isolates that had been previously frozen but failed to be revived from cryopreservation. The *T. gondii* genomes were obtained by enriching the parasite DNA in infected cells using the SureSelect Target Enrichment System, which enriches target genomes from non-live samples even though they contain minimal amounts of target genomes[25–27]. In summary, 3 out of the 7 frozen isolates (TgCatJp1, TgCatJp2, and TgMonkeyJp1) yielded high quality genomes with an average sequencing depth greater than 30. For the remaining four isolates (TgCatJp3, TgCatJp4, TgCatJp6, and TgCatJp7) partial genomes were resolved at high sequencing depth, with percentages of the genome providing high confidence data ranging from 74% to 10%, compared to the data obtained from directly sequencing DNA recovered from expanded isolates (Supplementary Table S1). Coverage plots were created for seven samples, revealing varying depth of coverage (Supplementary Fig. S2). The reads were mostly uniformly distributed across the entire chromosome, with a few regions showing significantly lower depth. These regions corresponded to low-complexity sequences in the reference genome. Strains obtained by the Target Enrichment were distinguished from isolates and used in subsequent analyses.

We created a Neighbor network based on genome-wide SNP data. Among the Japanese strains collected in this study, the most common genotype was clustered between HG2 and HG12 (3 out of 8: TgCatJpOk3, TgCatJpTy1/k3, and TgCatJpObi1), followed by two strains (TgBoJp1 and TgBoJp2) that were located close to the edges of HG2, and were highly similar to the reference type II ME49 genome (Fig. 1a and Supplementary Fig. S3a, b). An intra-group analysis of type II, HG12, and HG2-like Japanese strains showed that three strains—TgCatJpOk3, TgCatJpTy1/k3, and TgCatJpObi1—were distinct from both type II and HG12 strains (Supplementary Fig. S3c). Among them, TgCatJpObi1 specifically exhibited evidence of recombination with HG12. TgCatJpGi1/TaJ was found near the edge of HG3, and was highly similar to the reference type III VEG genome (Fig. 1a). Furthermore, TgCatJpOk4 appeared to be located midway between the branches of HG12 and HG13, suggesting that it is a recombinant strain. Interestingly, TgCatJp5 was adjacent to TgCtCo5 and TgRsCr01, which belong to HG15 of clade A found in South/Central America, but branched off from the root. Africa 4 (#20) cluster was located at the boundary of clade D. HG13 appears genetic recombination of HG2 and Africa 4 (#20).

Additionally, an analysis of the apicoplast genome (an organelle inherited maternally) identified 177 SNPs in the 35 Kbp genome. Most Japanese strains inherited the apicoplast from HG2 (Fig. 1b). HG13 haplotype was linked to TgCatJp5 and HG12 (RAY and B41) by a few SNPs, suggesting their kinships. ARI shared its genome with COUG (HG11) strain. The genetic distance between HG13 and HG2 suggests

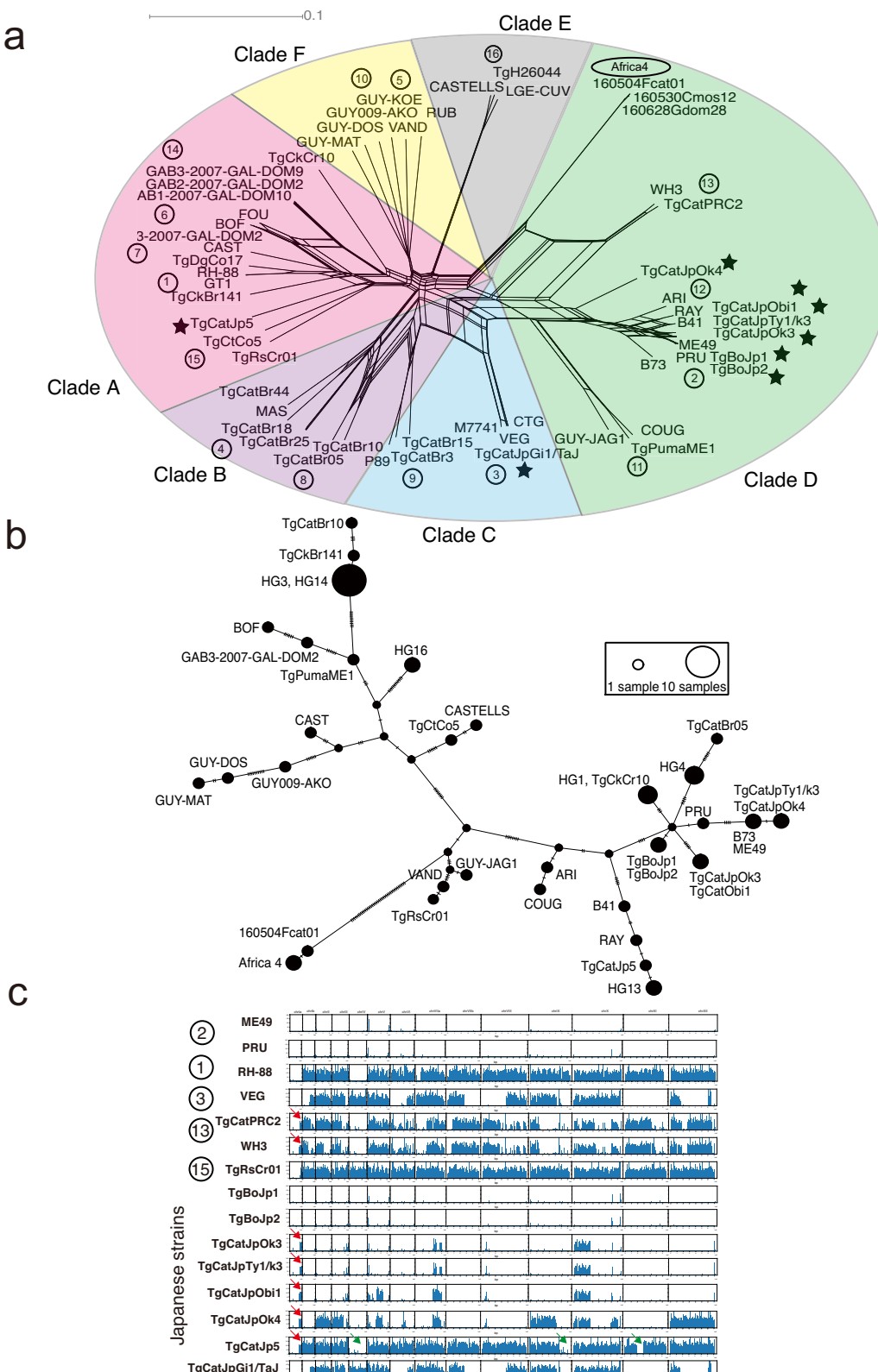

**Fig. 1 | Phylogenetic analysis of Japanese *T. gondii* based on genome-wide SNPs.**
**a** Neighbor-net analysis based on genome-wide SNPs (1,436,410 variant sites) from a total of 57 isolates, including 2–3 strains representing each of the 16 haplogroups plus three Africa 4 (#20), and 8 Japanese *T. gondii* strains. Stars indicate Japanese strains. The color wheels indicate major clades of *T. gondii*. Numbers in circles indicate haplogroups. **b** TCS network of apicoplast genome. **c** Pairwise SNP density plot. This plot shows a one-to-one comparison of SNP density with the reference strain (ME49). Each row represents the genome of one strain. Each column divides the chromosome based on length. Each vertical bar represents the number of SNPs within a 10 kbp window of the sequence. Red arrows indicate SNP patterns specific to HG13. Green arrows indicate the genome of HG2 within TgCatJp5.

that they diverged from a common ancestor and then followed different evolutionary pathways. Africa 4 (#20) lineage was distinct from other haplotypes, suggesting unique genetic characteristics.

To verify that TgCatJp5 and TgCatJpOk4 were not special strains resulting from accidental import or rare recombination, we performed Neighbor network analysis using whole genome SNPs for the 3 strains with high sequencing depth (>30) among the seven enriched samples. When restricted to SNPs at least four or more reads mapped for the other 4 strains, the Neighbor network revealed that TgCatJp4, TgCatJp7, and TgMonkeyJp1 belong to the HG2 subgroup (Supplementary Fig. S4a, c, e), while TgCatJp2, TgCatJp3, and TgCatJp6 are in the HG15 group (Supplementary Fig. S4a, b, d). TgCatJp1 was located between TgCatJpOk4 and HG3, appearing to be a distinct recombinant (Supplementary Fig. S4a). These results suggest that TgCatJp5 is a major group on the southern islands of Japan.

The genome-wide distance analysis indicates a significant amount of admixture in the Japanese strains (Fig. 1a). To investigate the admixture further, we created SNP density plots at every 10 kb and examined the recombination of haploblocks that represented differences from the reference type II (ME49) genomic sequence. We confirmed the admixtures of HG2 genome into HG1 (RH-88) and HG3 (VEG) as described previously[4] (Fig. 1c). As expected, the two strains (TgBoJp1 and TgBoJp2), which were part of HG2 (Fig. 1a), share a few very short blocks that are also found in PRU and a type II strain ME49 (Fig. 1c). On the other hand, the genomes of three strains (TgCatJpOk3, TgCatJpTy1/k3, and TgCatJpObi1), which were located between HG2 and HG12 in the Neighbor-net analysis (Fig. 1a), exhibited several large haploblocks that indicated more extensive admixture with other strains (Fig. 1c). The block at the end of chromosome Ia was common to all Japanese strains except for TgBoJp1, TgBoJp2, and TgCatJpGi1/TaJ (Fig. 1c red arrow). Furthermore, the SNP patterns on chromosomes V, VIIa, and X were quite similar to those in HG13 (Fig. 1c). In addition, TgCatJp5, which was found to be related to HG15, exhibited partial overlap (covering the full length of chromosomes Ia and IV, as well as chromosomes IX and XI) with the HG2 genome (Fig. 1c green arrow), suggesting the admixtures in Japanese strains. To confirm the admixture of TgCatJp5 and TgCatJpOk4, we overlaid their plots. We observed that haploblocks of TgCatJpOk4 different from the reference strain were almost identical to those of TgCatJp5 (Supplementary Fig. S5a). The SNP pattern of TgCatJpGi1/TaJ closely matched the variations representing HG3 (Supplementary Fig. S5b). Furthermore, to investigate the parent of TgCatJpOk4, we employed fineSTRUCTURE to generate a common ancestry matrix (Supplementary Fig. S6). The fineSTRUCURE analysis demonstrated a strong genetic exchange of TgCatJpOk4 with Japanese admixed HG2 and TgCatJp5, suggesting that TgCatJpOk4 is a progeny of the Japanese HG2 subgroup and TgCatJp5.

## TgCatJpOk4 inherited the type III ROP5 from the low virulent TgCatJp5, leading to its a highly virulent nature

TgCatJpOk4 showed high virulence in mice and microminipigs[23,28]. In contrast, the HG2 subgroup is known to be low virulent in mice. To analyze virulence of TgCatJp5, we analyzed TgCatJp5 in vivo and in vitro phenotypes[22,23,29]. Infection experiments revealed that all wild-type C57BL/6 mice survived until day 30 of infection, while all mice deficient in the IFN-γ receptor (*Ifngr1*[-/-] mice) succumbed (Fig. 2a), indicating that TgCatJp5 is a low virulent strain and is eliminated by an IFN-γ-dependent immune response. We further examined the recruitment of IFN-γ-inducible anti-*T. gondii* effectors such as IRGB6 and GBP2 to the parasitophorous vacuole membrane (PVM) in IFN-γ-stimulated mouse embryonic fibroblasts (MEFs) (Fig. 2b, and Supplementary Fig. S7a, b)[30]. An indirect immunofluorescence analysis exhibited that PVMs of TgCatJp5 parasites were coated with IRGB6 and GBP2 as highly as those of ME49 parasites (Fig. 2b and Supplementary Fig. S7a, b). In contrast, recruitment of these effectors on PVMs of RH

parasites was severely reduced (Fig. 2b and Supplementary Fig. S7a, b), consistent with in vivo virulence. Furthermore, since the recruitment of IFN-γ-inducible effectors on PVMs is shown to be inhibited by parasite virulence factors such as ROP5 and ROP18[31,32], we next analyzed types of ROP5 and ROP18 by in silico analysis (Fig. 2c, d). Referring to ROP5 copy number of archetypal I, II, and III, TgCatJpOk4 exhibited type III ROP5, as described previously[23] (Fig. 2c and Supplementary Fig. S7c). Moreover, TgCatJpOk4 possessed type II ROP18 allele (Fig. 2d). We found that TgCatJp5 carries a low copy number, highly active type III ROP5 (ROP5[III]) allele, and a low activity type III ROP18 (ROP18[III]) allele with some deletion in the UPS region (Fig. 2c, d and Supplementary Fig. S7c). The Japanese HG2 (TgCatJpOk3) possessed inactive ROP5 (ROP5[II]) and active ROP18 (ROP18[II]). Based on the results so far, the low virulent strains were attributed to having inactive ROP5[II] in the Japanese HG2 and inactive ROP18[III] in TgCatJp5. However, TgCatJpOk4 had active ROP18[II] and active ROP5[III]. Altogether, the likely explanation for the high virulence of TgCatJpOk4 is due to the inheritance of ROP18[II] from the HG2 (such as TgCatJpOk3) and ROP5[III] from TgCatJp5 lineage (Fig. 2E), as was previously observed for the S23 recombinant F1 progeny that became mouse-virulent after inheriting the ROP18[II] allele x ROP5[III] allele from mouse-avirulent parent strains ME49 (type II) and CTG (type III), respectively[33].

## Visualization of the population genome structure using POPSICLE reveals HG13 haploblocks in the HG12 genome

Although SNP density plots can identify regions that differ from reference strains (Fig. 1c), they cannot determine the ancestors of non-reference haploblocks. Therefore, we performed cluster analysis to infer ancestral structure using programs such as ADMIXTURE and fastSTRUCTURE, both of which are used in population genetics[34,35]. These tools, based on the model wherein individuals are genetically derived from multiple ancestral populations, illustrate the 'mixture' of genetic ancestors within a population. The genetic composition of each individual is represented by the proportion of genetic contributions, represented as different colors, with each color corresponding to a specific ancestor. To investigate tools capable of reproducing the local admixture observed in *T. gondii* genomes, we used the 62 global *T. gondii* genomes published in a previous study[4]. The choice of the optimal "K", which is obtained by finding the population size that minimizes the cross-validation error in ADMIXTURE, while in fastSTRUCTURE it is the value that maximizes the marginal likelihood. For a given "K" choice, ADMIXTURE (K = 8) and fastSTRUCTURE (K = 6) show at least one unique color for each clade (Supplementary Fig. S8a−d). Both programs captured the admixture of strains reported to be recombinants (TgDgCo17 and B73)[14], but did not resolve the admixture of the HG2 genome in HG1 (RH-88) and HG3 (VEG), as shown in the SNP density plot (Fig. 1c and Supplementary Fig. S8b, d) or in previous studies[4,14]. In addition, the pattern of P89, which shares partial ancestry with HG3[14], could not be resolved (Supplementary Fig. S8b, d). Furthermore, there was inconsistency in the portion of the HG2 genome in HG12 (ARI) shown in the previous study (Supplementary Fig. S8b, d)[4]. These results imply a limitation in the ability of these tools to accurately decipher mixed ancestry as established in earlier research[4,14].

To provide a comprehensive and visual representation of *T. gondii* strain diversity, we used the POPSICLE software, which divides the genome into non-overlapping sliding windows of user-defined size (here 10 kb) and assigns clades to local regions. Since POPSICLE has already been used in another study[24] but has never been used in *T. gondii* research, we evaluate its performance using the 62 global strains described above. The maximum value of "K = 14" was chosen for POPSICLE, using the Dunn index as an index (Supplementary Fig. S9a). The value of "K" was consistent with the number of clusters as indicated by fineSTRUCTURE (Supplementary Fig. S6). To compare the POPSICLE plot with ADMIXTURE and fastSTRUCTURE, it was displayed in the

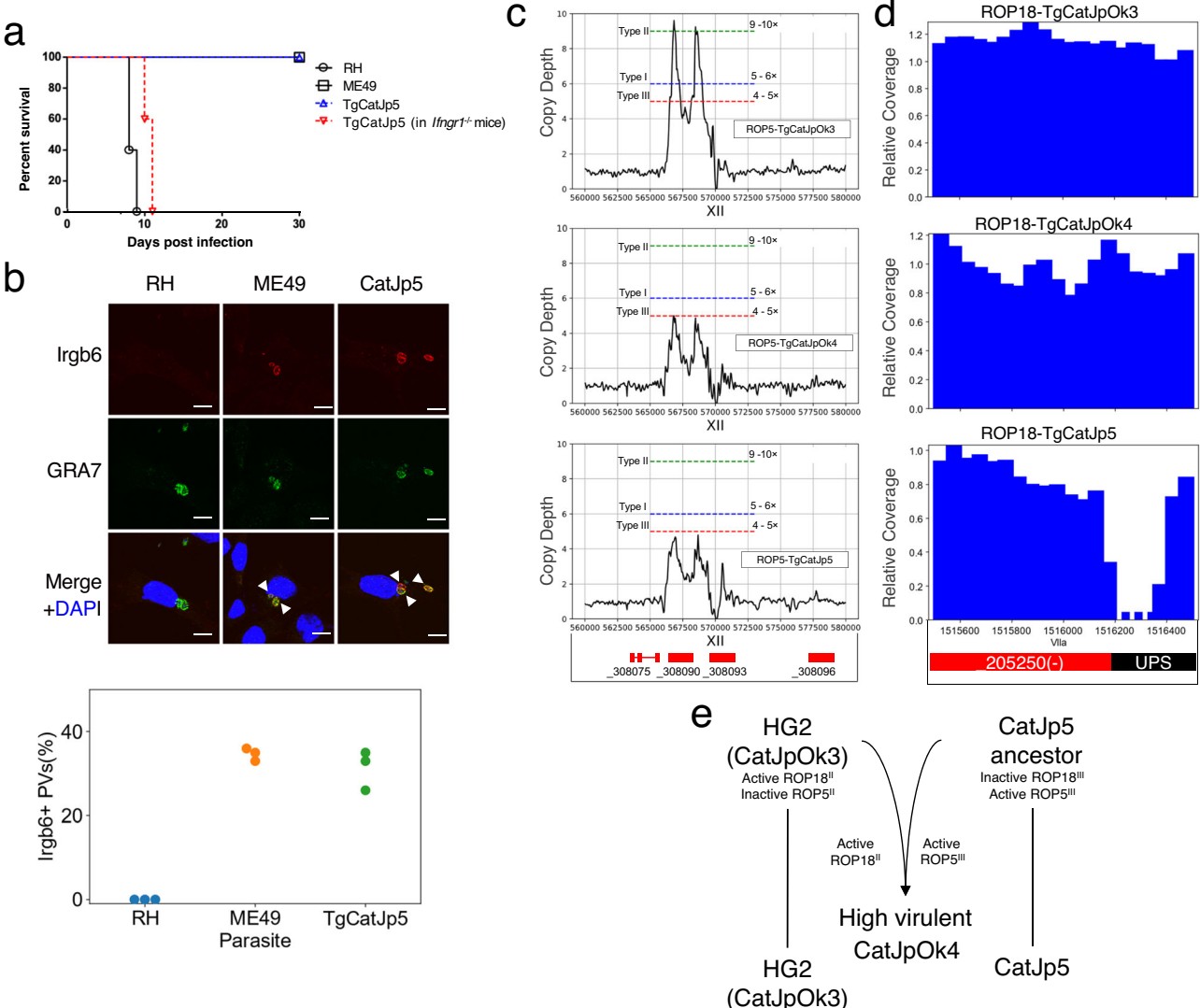

**Fig. 2 | In vivo and in vitro characterization of TgCatJp5. a** Evaluation of in vivo virulence in mice. The pathogenicity of TgCatJp5 in mice was evaluated by inoculating 1000 tachyzoites in C57BL/6 ($n = 5$, each) or $Ifngr1^{-/-}$ mice ($n = 5$) in parallel with RH and ME49. All mice inoculated with RH died by day 9 after inoculation, while mice inoculated with ME49 and TgCatJp5 all survived to day 30 post-infection. $Ifngr1^{-/-}$ mice succumbed to infection completely by day 11 post-TgCatJp5 infection. Source data are provided as a Source Data file. **b** Accumulation of Irgb6 on PV membranes. Accumulation of Irgb6 into PV membrane was tested by infecting IFN-γ-stimulated mouse embryonic fibroblasts (MEFs) with the parasites HG1 (RH), HG2 (ME49), and TgCatJp5. Two hours later, cells were fixed, and immunofluorescence staining was performed using GRA7 (green), Irgb6 (red), and DAPI (blue). The scale bar represents 10 μm. $N = 3$ biologically independent samples, $n = 100$ cells were examined over experiment. Arrowheads indicate Irgb6+ parasitophorous vacuoles. Source data are provided as a Source Data file. **c** Copy number variation of ROP5. Sequencing depth at every 1000 bp was calculated by tinycov. CNV was calculated by dividing the maximum sequencing depth of the ROP5 gene region by the average sequencing depth of chromosome XII. Dashed lines represent estimated copy depth of the archetype strains (blue, type I/RH-88; green, type II/ME49; red, type III/VEG). **d** Mapping status for the UPS region of ROP18. Sequencing depth was calculated by tinycov every 50 bp, showing the 1,515,500–1,516,500 bp range of chromosome VIIa containing the UPS region of ROP18. **e** Relationship between TgCatJp5 and TgCatJpOk4 assumed from the allele exchange of ROP5 and ROP18.

same style as these tools (Supplementary Fig. S8b). An original POPSICLE plot is shown (Supplementary Fig. S9c). POPSICLE resolved the admixture of the HG2 genome into HG1 and HG3 (Supplementary Fig. S9b, c). In addition, the ancestral admixture in P89 and ARI was consistent with previous studies[4,14] (Supplementary Fig. S9b, c). In contrast, the results of "K = 14" in ADMIXTURE and fastSTRUCTURE segmented each HG, but did not seem to resolve the admixture of genetic ancestry within the population (Supplementary Fig. S8b, d). Taken together, these results suggest that POPSICLE more closely conform to prior assumptions. These three in silico methods are powerful tools, but each cannot perfectly mimic the underlying population and genetic processes. This underscores the importance of using multiple methods to cross-validate findings in population genetics.

The feature of POPSICLE is to visualize genomic structures by assigning unique colors to each ancestral lineage. While it is technically feasible to include all 57 strains in the POPSICLE analysis, as the number of ancestral lineages increases, the colors become more subdivided, leading to decreased visibility. To mitigate this issue and produce clearer plot, we strategically excluded ancestral lineages with low relevance to Japanese strains from our analysis. The selection was based on the results of POPSICLE, created using various patterns, including data substitution and resampling (Supplementary Fig. S10a–c). We then used the minimal set of ancestral populations necessary to depict the genomic structure of the *T. gondii* population in Far East Asia. In this study, we also employed the cluster evaluation metric, Dunn index[36] to support our estimate calculation. Although the

maximum value was reached at k = 12 for an ancestral population size between 2 and 15, we chose k = 10 for the analysis due to the subdivision of the three HG15 lineages (TgCkCr10, TgRsCr01, and TgCtCo5) at k = 11 and 12 (Supplementary Fig. S10d–f). Using POPSICLE, we calculated the number of clades present within each 10 kb sliding window. Local clades were assigned different colors and painted across the genome to resolve ancestry (Fig. 3a). To verify that the POPSICLE software was performing as expected, we analyzed genome-data derived from archetypal I, II, and III and 2 recombinant lines (B73 and TgDgCo17), where admixture among ancestors is well-documented (Fig. 3b)[14,37]. The inner frames of the HG3 strains (CTG, VEG, and TgCatJpGi1/TaJ) are colored brown (Fig. 3b). The outer frames show a mixture of yellow and brown, and the center frames represent the percentage of the ancestral lineage comprising HG3 (60% brown, 40% yellow). It was previously established that HG3 a recombinant line, was the result of a few crosses between the HG2 ancestral lineage with that of another ancestral lineage, referred to as β[37]. Thus, the ancestor of HG2 and the ancestor of β are drawn as yellow and brown in this plot, respectively (Fig. 3b). Here we name the unique ancestors of HG2 and HG3 as HG2$^{yellow}$ and HG3$^{brown}$, respectively. As observed in the SNP density plot (Fig. 1c), HG2 strains (ME49 and PRU) appeared as 99% of HG2$^{yellow}$ with minimal admixture (Fig. 3a). HG1 displayed admixture of 10% of HG2$^{yellow}$ in addition to 50% of HG1$^{navy}$ and 40% of HG6$^{deep green}$, which are the unique ancestors of HG1 and HG6, respectively (Fig. 3a, b). Moreover, as previously described[38,39], B73 and TgDgCo17 appeared to be F1 progeny of HG2 and HG3 and HG1 and HG3, respectively. Similarly, B41 has previously been suggested to be a recombinant progeny between HG2 (ME49) and HG12 (RAY), and our result was consistent with the previous analysis[14] (Fig. 3c). Furthermore, we found that the POPSICLE analysis classified the HG13 strains (TgCatPRC2 and WH3) as a mosaic including 35% HG2$^{yellow}$ (Fig. 3d), which was consistent with a previous study that HG13 shares genomic regions with ME49 (~40%)[4]. Thus, these results indicated that POPSICLE could depict recombination and admixture between known haplogroups, and furthermore reveal the extent and specific location of admixture within the genomes. Notably, the small portion of HG13$^{red}$ was also conserved in HG12 (ARI, RAY, B41) (Fig. 3d), suggesting that HG13 and HG12 share the Far East Asian ancestral HG13$^{red}$ lineage. Moreover, approximately 34% of the HG13 genome was introgressed with the Africa4$^{blue}$, while only less than 0.1% of the HG12 genome showed such admixture.

### Identification of shared ancestral haploblocks between TgCatJpOk4 the North America-specific HG12

Next, we focused on the population structure of the Japanese *T. gondii* strains based upon the POPSICLE analysis (Fig. 4a). TgBoJp1 and TgBoJp2 showed close similarity to HG2 (ME49 and PRU) (Fig. 4a), which were consistent with the SNP density plot (Fig. 1c). As the SNP fingerprint predicted the haploblock of HG13$^{red}$ in TgCatJpOk3, TgCatJpTy1/k3 and TgCatJpObi1 (Fig. 1c), the POPSICLE analysis visualized the extent and position of the HG13$^{red}$ haploblocks in the genome of in TgCatJpOk3, TgCatJpTy1/k3 and TgCatJpObi1, suggesting admixture of HG13$^{red}$ into the Japanese *T. gondii* strains. Notably, TgCatJpOk4 was also assigned a unique ancestor (magenta), which hereafter we named JpOk4$^{magenta}$ (Fig. 4a), and other patterns supported the prediction that TgCatJpOk4 is a mixture of TgCatJp5 and the Japanese HG2-subgroup such as TgCatJpOk3 (Figs. 2e and 4a). When we analyzed the genomic localization of JpOk4$^{magenta}$ in other strains, the haploblock was highly conserved on chromosome V for the three HG12 strains (ARI, RAY, B41), in addition to a Japanese HG2 subgroup strain (TgCatJpObi1) (Fig. 4b). We then performed Neighbor-net analysis using the concatenated sequence (1.22 Mbp) of the JpOk4$^{magenta}$ haploblocks identified throughout the chromosome (Fig. 4c). Surprisingly, we observed that this magenta hue shared a common root with HG11 (COUG, GUY-JAG1, TgPumaME1) in clade D

but that it had accumulated a significant number of SNPs (Fig. 4c). This data suggests that we identified traces of an unknown ancestral lineage that characterizes HG12 and the Japanese HG2 subgroup by comparing the hue determined in the POPSICLE with other lineages. To ensure reproducibility, and to include strains that were not present in the POPSICLE analysis, we conducted a phylogenetic analysis of the same region (magenta of chromosome V). We added 3 strains (TgCatJp1, TgCatJp2, and TgMonkeyJp1) that had an average sequencing depth > 30 in the target enrichment. TgMonkeyJp1 belonged to the HG12 cluster, along with TgCatJpOk4 and TgCatJpObi1 (Fig. 4d). Taken together, these results indicated that HG12 and some Japanese strains share the same ancestral lineage depicted by JpOk4$^{magenta}$ (Fig. 4e).

### Identification of haploblocks of the TgCatJp5 ancestral lineage found in the genomes of several HGs of Central and South American *Toxoplasma*

Not only the unique ancestral lineage from TgCatJpOk4, POPSICLE identified another unique ancestry (depicted by the color cyan), which hereafter we call Jp5$^{cyan}$, from TgCatJp5 (Figs. 3a and 4a). POPSICLE determined the population genetic structure of TgCatJp5 as a complex admixture involving HG1$^{navy}$, HG2$^{yellow}$, HG3$^{brown}$, HG6$^{deep green}$, HG13$^{red}$ and JpOk4$^{magenta}$. It was noteworthy that TgCtCo5 and TgRsCr1 strains of HG15 as well as the TgCatJpOk4 strain shared Jp5$^{cyan}$, suggesting that these four strains share a common ancestral lineage (Fig. 3a, intermediate plot). Therefore, to investigate the history of cyan, we focused on genomic regions where cyan hues were assigned in TgCatJp5 and TgRsCr01 or TgCtCo5 or both (chromosomes III (0.54 Mbp), VIIa (1.68 Mbp), X (0.92 Mbp) (Fig. 5a, Supplementary Fig. S11a, b). Interestingly, the cyan haploblock clearly separated TgCtCo5, TgRsCr01, and TgCatJp5 from other clade A lineages (HG1 and HG6) (Fig. 5b, Supplementary Fig. S11c, d). Distance analysis using more strains showed that the cyan region on chromosome VIIa formed a cluster with TgCatJp2, TgCatJp5, and TgRsCr01 (Fig. 5c). In contrast, in other regions on chromosomes II, III and XII, where the cyan haploblocks were well conserved, they formed long branches that branched off from the root and there was no clear evidence of recombination with other HGs (Supplementary Fig. S12a–f). Additionally, to explore the possibility of shared haploblocks between TgCatJpOk4 and TgCatJp1, we generated a POPSICLE plot that includes TgCatJp1 (Supplementary Fig. S10d). This revealed that the TgCatJp1 genome serves as an intermediate between TgCatJp5 and TgCatJpOk4, suggesting a kinship with TgCatJpOk4. Furthermore, the additional POPSICLE analysis identified haploblocks such as HG1$^{navy}$, HG3$^{brown}$, and HG6$^{deepgreen}$, which deviate from HG2 in TgCatJp1 and TgCatJp2 (Supplementary Fig. S10d). This indicates that hybrid capture can capture sequences from different HGs for ancestor analysis, even with ME49-based probes.

Since all strains of clade F were from French Guiana in South America, we postulated that the cyan haploblocks would be found in other South American strains. To test this hypothesis, POPSICLE analysis was performed to visualize the population genetic structure that included South American lineages belonging to HG5 and HG10 such as VAND, RUB, GYU-KOE, GYU-MAT, GUY-DOS and GUY-009-AKO strains (Fig. 5d). The HG5 and HG10 genomes were resolved as a mosaic pattern with a major ancestry (khaki) alongside many other colors. Although the locations were on different chromosomal coordinates (5−32 blocks of 10 kbp window size), all HG5 and HG10 strains exhibited small blocks of Jp5$^{cyan}$ (Fig. 5e and Supplementary Fig. S13), suggesting potential shared ancestry (Fig. 5f).

### Evaluation of the genetic distance between the HG2, HG13, and Africa 4 genomes

To characterize the Africa4$^{blue}$ genomic region, which constitutes approximately 30% of the HG13 genome, we focused on the Africa4$^{blue}$ region in both TgCatPRC2 (HG13) and 160504Fcat01 (Africa 4)

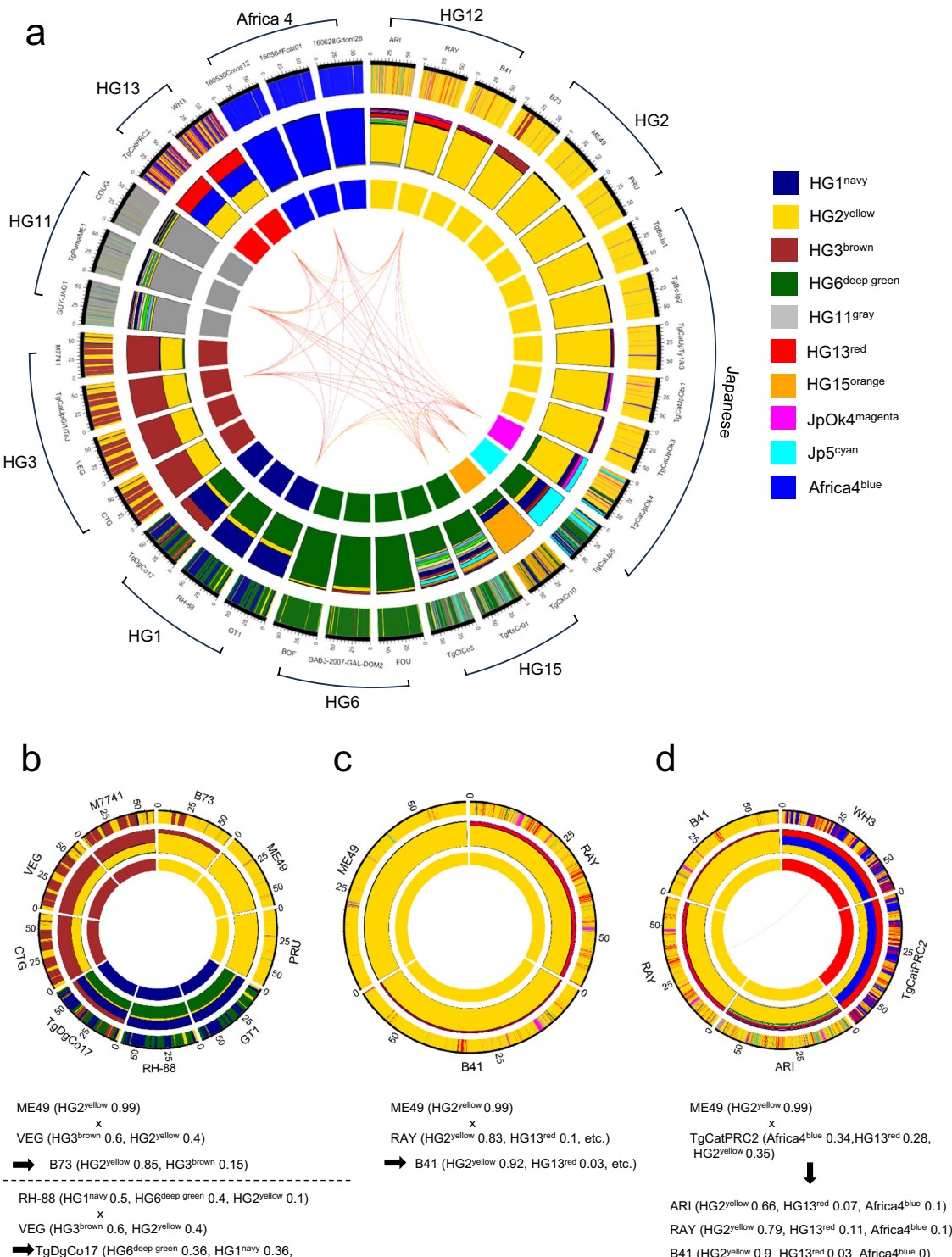

**Fig. 3 | *T. gondii* ancestry analysis using POPSICLE. a** Clustering analysis of the *T. gondii* genome from POPSICLE using k = 10 for population genetic structure and admixture. The haplotype plot with 34 strains identified eight "major ancestral lineages" corresponding to existing genotypes (inner circle plot: Navy (HG1), yellow (HG2), brown (HG3), deep green (HG6), gray (HG11), red (HG13), orange (HG15), and blue (Africa 4)). Two 'individual ancestors' of color were also resolved in the Japanese population (inner circle plot: magenta (TgCatJpOk4), cyan (TgCatJp5)). Hues did not appear in the inner circles (e.g., light green) indicating that they are unresolved in the present dataset. For each sample, we assigned a hue representing the "type" of gene present and assembled an ancestral haplotype block. The middle circle represents the relative proportion of each gene ancestor in each genome, expressed as a hue. The outer circle represents the genome-wide local admixture profile of each isolate in a 10 kb sliding window. By arranging these hue patterns in chromosomal order (chromosomes Ia to XII), the genome-wide admixture pattern was resolved. **b** POPSICLE plot from which HG1, HG2, and HG3 strains were extracted. **c** POPSICLE plot from which ME49 (HG2), RAY (HG12), and B41 (HG12) were extracted. **d** POPSICLE plot from which HG12 and HG13 strains were extracted.

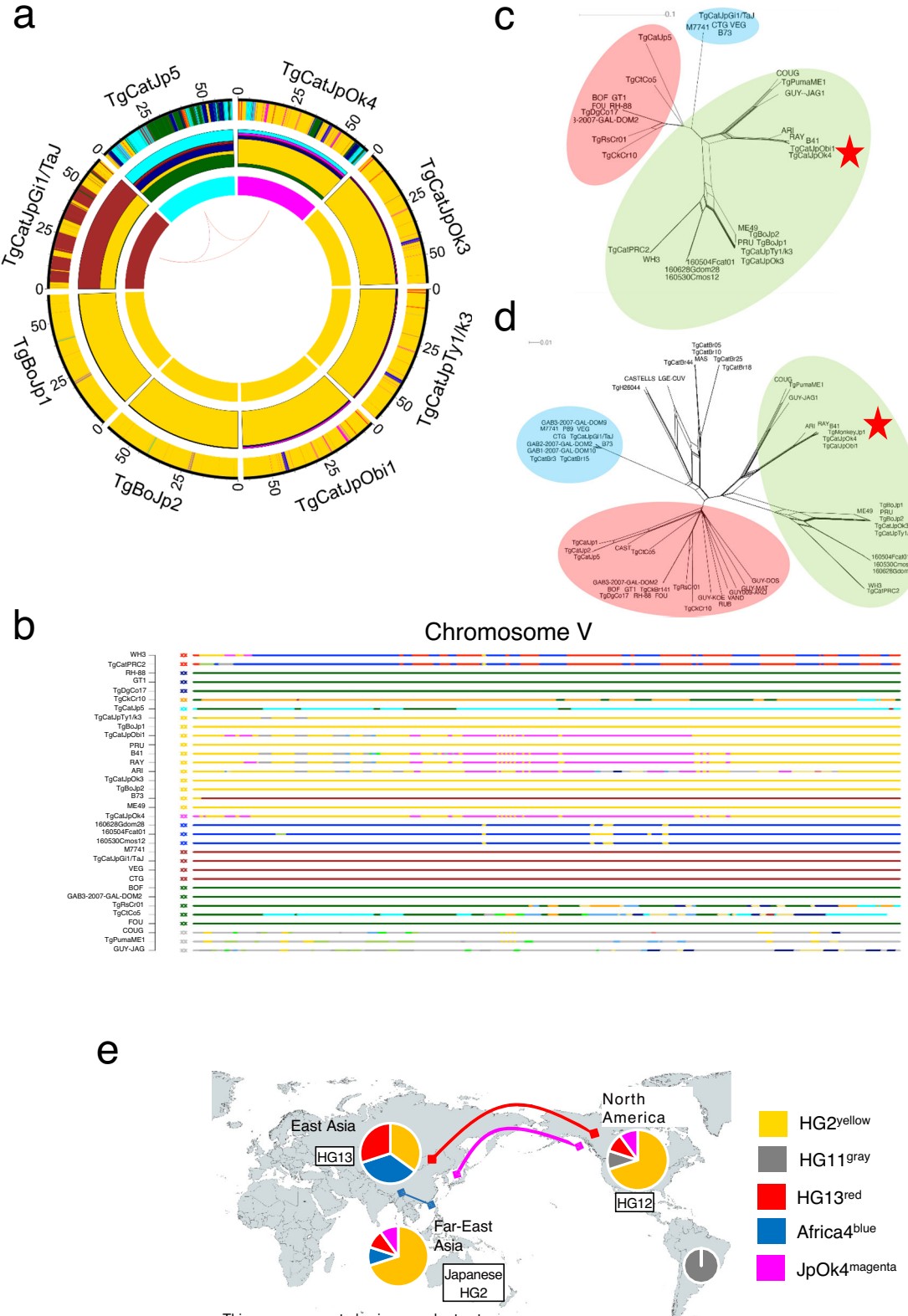

**Fig. 4 | Genomic regions that characterized the HG12 and Japanese HG2 subpopulations. a** POPSICLE plot from which Japanese strains were extracted. **b** Chromosome painting of chromosome V. **c, d** Neighbor-network based on the concatenated sequence of the regions assigned magenta in TgCatJpOk4 and RAY in **a**. The colored wheels indicate the major clades of *T. gondii*. **c** Network by the strains used in the POPSICLE analysis. **d** Network by all HGs including three strains (TgCatJp1, TgCatJp2, TgMonkeyJp1) revived by Target Enrichment. **e** A diagram map showing the ancestral lineage shared by East Asian and North American *T. gondii*. Each pie chart represents the rough ancestral structure of HG12, HG13, and Japanese HG2. The curve connecting the two regions indicates that they share an ancestral lineage with each other. The map was created using mapchart.net.

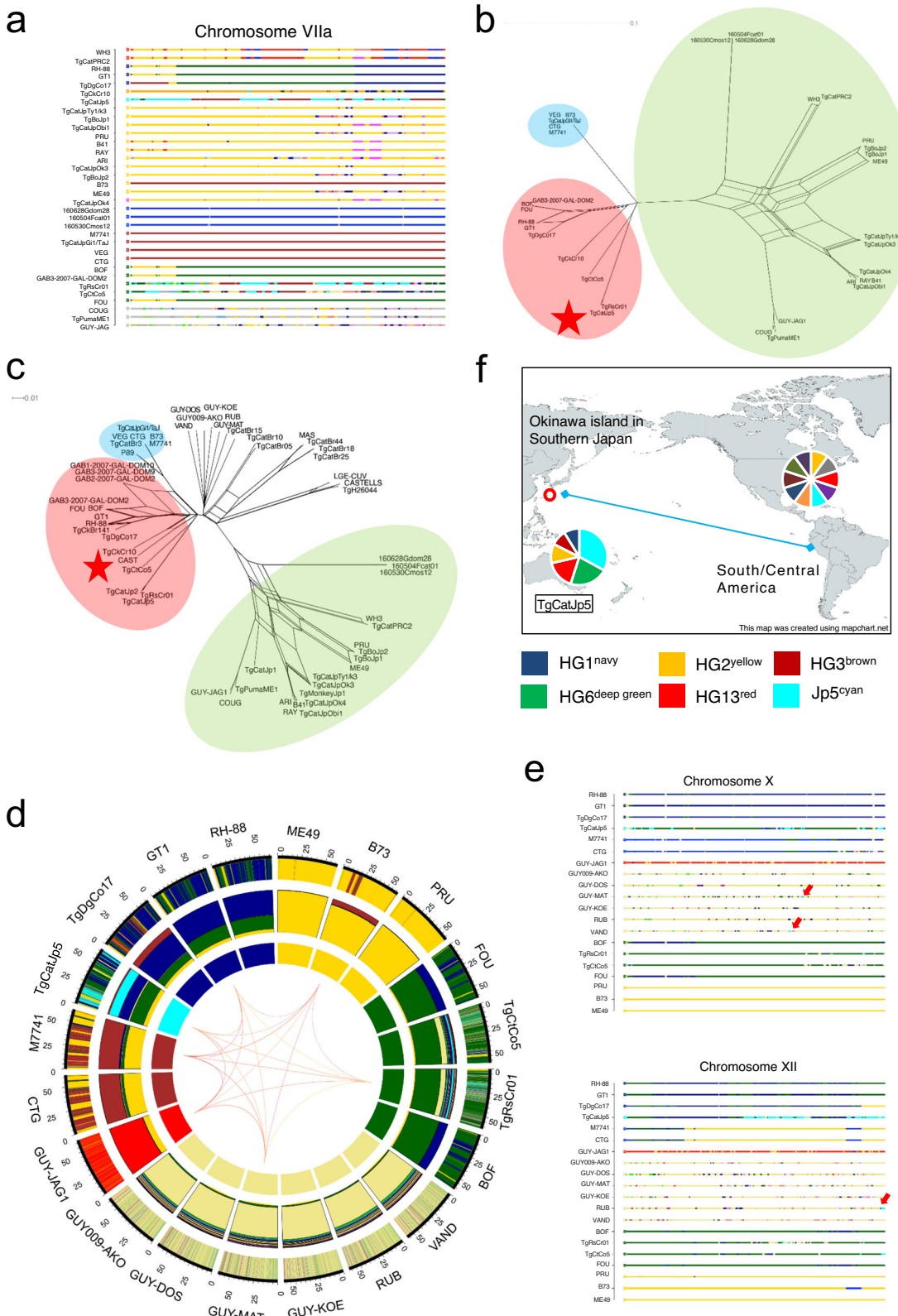

**Fig. 5 | Classification of ancestral lineages characterizing the Japanese *T. gondii* population. a** Chromosome painting of chromosome VIIa. **b, c** Neighbor-network based on the concatenated sequence of the regions assigned cyan in TgCatJp5 and TgRsCr01 in **a**. The colored wheels indicate the major clades of *T. gondii*. **b** Network by the strains used in the POPSICLE analysis. **c** Network by all HGs including three strains (TgCatJp1, TgCatJp2, TgMonkeyJp1) revived by Target Enrichment. **d** POPSICLE plot was made with the addition of HG5 and HG10. **e** Chromosome painting of chromosome X and XII. **f** A diagram map showing the ancestral lineage shared by East Asian and Central and South American *T. gondii*. Each pie chart represents the rough ancestral structure of TgCatJp5 and highly recombinant South American strain. The curve connecting the two regions indicates that they share an ancestral lineage with each other. The map was created using mapchart.net.

(Fig. 6a). A neighbor-net analysis showed that the Africa4[blue] region in HG13 is closely related to the Africa 4 (#20) genome (Fig. 6b). Furthermore, we focused on the genomic region where color hues were assigned as HG2[yellow] in ME49, HG13[red] in TgCatPRC2, and Africa4[blue] in 160504Fcat01. In these unexchanged regions, the strong recombination relationship previously observed between HG13 and Africa 4 (#20) in the whole genome was resolved (Figs. 1a and 6c). In this region (17.04 Mbp), we identified 128,689 SNPs in 160504Fcat01 (Africa 4) compared to 82,500 SNPs in TgCatPRC2 (HG13) (Supplementary Table S3). This is consistent with the apicoplast genome, suggesting a more recent divergence between HG2[yellow] and HG13[red] than with Africa4[blue]. To investigate the extent to which the Africa4[blue] genome is conserved in lineages other than HG13, we examined the proportion of haploblocks within each 10-kbp-sliding window. The admixed Japanese HG2 strains had 1.9–2.1%, while only very small fragments (0.1–0.3%) were observed in HG11, HG12, and HG15 (Supplementary Table S4). Altogether, these results suggest that Africa 4 genome contributes to the *T. gondii* populations of HG13 and Japan, as well as to the Indian Ocean coastal region where Africa 4 (#20) lineage is frequently distributed (Fig. 6d).

## Discussion

Previous work was performed to describe the population genetic structure of *T. gondii* at whole genome resolution. However, since number of *T. gondii* strains from Africa and Asia are insufficient, their high-resolution genetic structures remain unclear. In particular, Asia was underrepresented as only one isolate, TgCatPRC2, has been included in all previous large-scale population genomic studies[4,14,15]. Our current study focuses on *T. gondii* in Asia, using high resolution genomic data obtained from eight isolates from Japan, two Chinese HG13 strains, and 7 genomes resurrected from non-viable isolates using a Target Enrichment Sequencing approach.

We demonstrate that the most prevalent genotype in Japan was the cosmopolitan HG2 archetypal lineage. However, a significant number of recombinants were also resolved that shared ancestry with HG12 and HG13 strains. The identification of HG13[red] admixture as well as HG12 haploblocks from North America into HG2 present in Far East Asian isolates strongly indicates a link between *T. gondii* in Far East Asia and North America, which is consistent with a recent report indicating that HG12 and HG13 share a common apicoplast ancestor[15]. Additionally, our present study identified JpOk4[magenta], a genomic region on chromosome V (1.28–2.4 Mbp), located near the genome of HG11 (HG11[gray]). This region overlaps with a segment (1.25–2.1 Mbp) previously proposed to be similar between HG11 and HG12 in the haplotype map by Minot et al.[14] Given that the region is closer to HG11 than its sister lineages HG2, these results indicate that JpOk4[magenta] is the region separating and characterizing HG12 from other HGs. The inclusion of genomic data from more Far East Asian and North American strains in the future, and the discovery of such ancestral lineages may provide valuable insights into understanding the evolutionary origin and history of *T. gondii* clade D.

Our investigation of the 9 isolates in Okinawa Island of southern Japan revealed that closely related strains of HG15 are also prevalent on the island. A previous study classified TgCatJpOk4 as HG2 by a low-resolution genotyping[23]. However, our high-resolution genome-wide analysis suggested that TgCatJpOk4 was produced by crossing the TgCatJp5 lineage, which is closely related to HG15, with a local HG2 subgroup such as TgCatJpOk3[23].

In fact, the unique Jp5[cyan] ancestral lineage was found not only in the recombinant progeny TgCatJpOk4 isolate, but also in the genomes of HG15 (TgCtCo5: Colombia in the northern tip of South America and TgRsCr01: Costa Rica in Central America), HG5 and HG10 from French Guiana. The genome structures of the South/Central American strains analyzed using POPSICLE (including TgRsCr01, TgCtCo5, and GUY-JAG1: French Guiana) showed a complex mosaic pattern, including colors not identified as the major ancestor. Overall, the *T. gondii* strains in South America showed no obvious dominant genotypes and have a population structure dominated by recombination, which differs significantly from other geographical regions. The result of finding Jp5[cyan] haploblocks in the South/Central American strains was surprising. The Jp5[cyan] haploblock was present in a strain found on Okinawa Island, a small island in Japan, geographically isolated from the rest of the world, and far from South America. This data may suggest that the cyan block, that has diverged from other strains in Clade A, has been maintained on the remote island in Japan and represents a unique common ancestry that existed in both South/Central America and Far East Asia, prior to the purification and fragmentation of this ancestral lineage by extensive recombination common among *T. gondii* populations in South/Central America. Hence, the Jp5[cyan] lineage from the POPSICLE analysis appears to identify a haploblock causally linking *T. gondii* in Far East Asia with South/Central America. Previous reports showed isolations of South American and African *T. gondii* isolates in Asia. Four strains found in dogs in Vietnam correspond to RFLP#18, which includes three strains from Colombia and two from China[40]. Furthermore, HG13, which is predominant in China, has been sporadically identified in the North, Central, and South America (one strain from Colombia, one from Mexico, and one from the United States) and Southeast Asia (one from Sri Lanka and four from Vietnam). The North American wild genotype RFLP#5 has also been reported in China. Additionally, Africa 4 (#20), which differs from HG13 by only one allele of the RFLP markers used, has been identified in East Africa (Egypt and Ethiopia), the Emirates and Sri Lanka, further supporting the history of intercontinental migrations of *T. gondii* in the recent past[12,41–43]. Once the whole genomes of these strains become available, it should become possible to elucidate detailed migration routes and histories between Far East Asia and South America and precisely how they are related evolutionarily with Japanese and Chinese lineages.

Notably, TgCatJpOk4 not only demonstrated significant virulence in mice comparable to type I but it also caused death in 3 out of 5 mini-pigs[28]. The virulence in mice was attributed to a combination of ROP5[III] and ROP18[II], which inhibited host immunity[31,32,44]. However, all the mini-pigs survived when inoculated with the type I RH strain, suggesting that TgCatJpOk4 contains other host-specific virulence factors, that make it more virulent in pigs than one of the most virulent strains thus far described[23,28,45]. TgCatJp1 was also considered to be a genetic cross between TgCatJp5 and HG2 such as TgCatJpOk3. The presence of TgCatJp1 indicates that the recombinant, like TgCatJpOk4, has been repeatedly produced, suggesting that such hybrid *T. gondii* genomes are endemic in this area. Although the pathogenicity of HG15 to humans is unknown, our study suggests that highly pathogenic hybrids could emerge in Okinawa, in a situation analogous to South/Central America, where hybrid strains that cause toxoplasmosis in immunocompetent humans exist[46–49]. There is no significant difference in the seropositivity rate of *T. gondii* in fattening pigs (6–7 months) between Okinawa and mainland Japan[50,51]. However, out of the 166 cases of symptomatic porcine toxoplasmosis diagnosed during slaughter inspections in Japan over the past five years (2018–2022), 165 cases were identified in Okinawa. Post-slaughter inspections revealed swelling, hemorrhage, and necrosis of internal lymph nodes, as well as hemorrhage of the liver in most affected pigs. Furthermore, although the national average rate of *T. gondii* antibody positivity among Japanese is approximately 6%, which is low compared to other countries' rates, antibody possession in Okinawa has been found to be as high as 12.9% among humans[18,52]. The higher seropositivity rates and number of symptomatic swine could be explained by socio-economic and cultural differences between Okinawa and the rest of Japan. However, highly pathogenic hybrids such as TgCatJpOk4 could also play a significant role. Consequently, isolating and analyzing *T. gondii* from symptomatic swine in Okinawa is needed to enhance our understanding of the risks to public health.

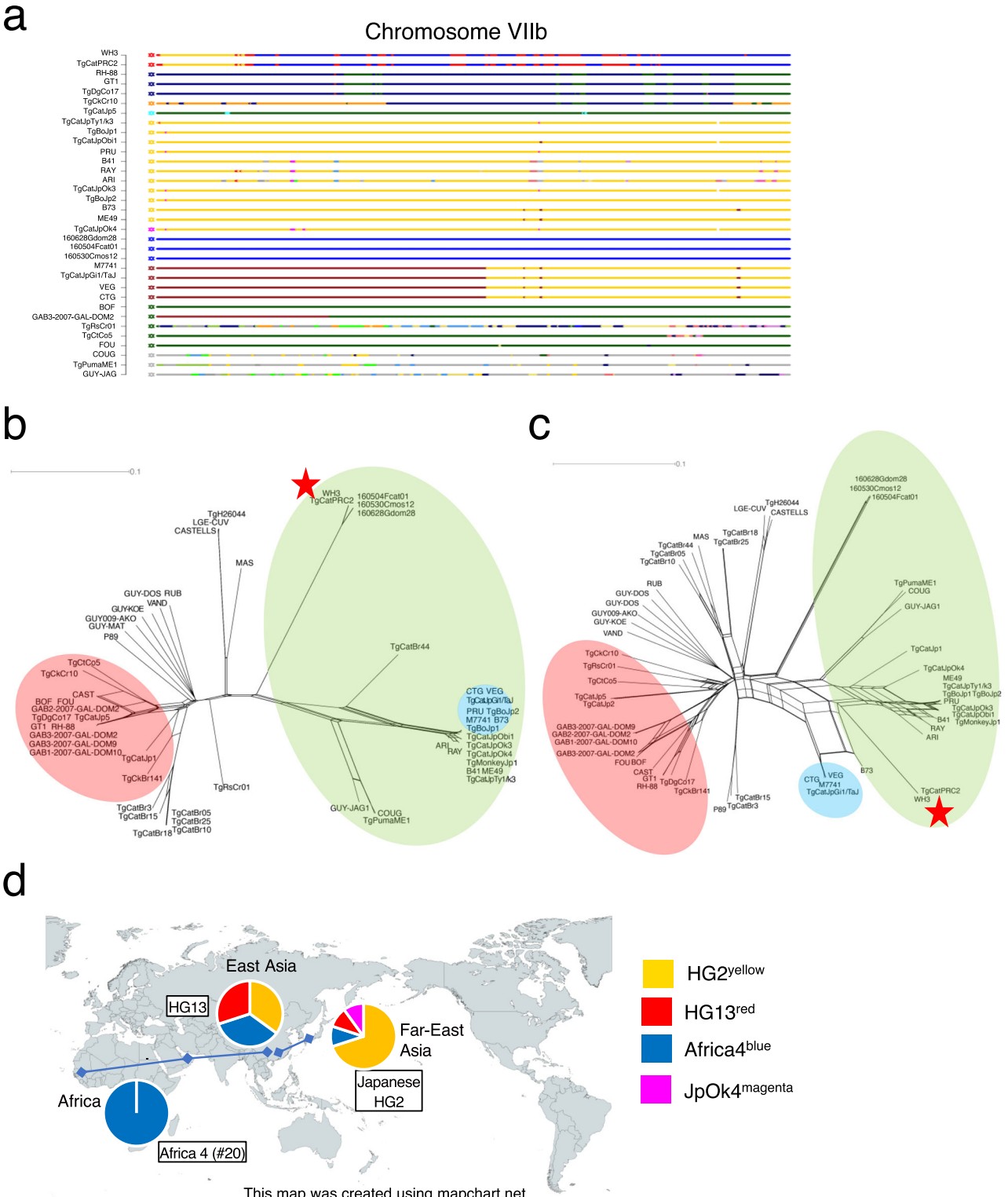

**Fig. 6 | Evaluation of the genetic distance among the HG2, HG13, and Africa 4 genomes. a** Chromosome painting of chromosome VIIb. **b** Neighbor-network based on the concatenated sequence of the regions assigned blue in TgCatPRC2 and 160504Fcat01 in **a**. The colored wheels indicate the major clades of *T. gondii*. Network by all HGs including three strains (TgCatJp1, TgCatJp2, TgMonkeyJp1) revived by Target Enrichment. **c** Neighbor-network based on the concatenated sequence of the regions assigned yellow in ME49, red in TgCatPRC2, and blue in 160504Fcat01 in Fig. 3a. The colored wheels indicate the major clades of *T. gondii*. Network by all HGs including three strains (TgCatJp1, TgCatJp2, TgMonkeyJp1) revived by Target Enrichment. **d** A diagram map showing the ancestral lineage shared by East Asian and Africa 4 (#20) *T. gondii*. Each pie chart represents the rough ancestral structure of Africa 4 (#20), HG13, and Japanese HG2. The curve connecting the two regions indicates that they share an ancestral lineage with each other. The map was created using mapchart.net.

**Fig. 7 | Journey of Clade D *T. gondii* considering recombination.** A hypothetical diagram illustrating the historical migration events of Clade D *T. gondii*. The sequence of events from 1 to 5 corresponds to specific genetic recombination episodes. The red arrows represent the directions of genetic admixture. Asterisks on HG13 and HG12 indicate that these genetic lineages had ancestral-type genomes, which differed from their current genome.

The type II, estimated to have appeared about 26,000 years ago[15], was selected for by the domestication of the Libyan wildcat in the Fertile Crescent with the advent of agriculture about 10,000 years ago, and subsequently spread to Europe and Asia[53]. It is known that agriculture also began in China and parts of Africa, suggesting that the ancestral lineages of Africa 4 (#20) and HG13 predated the arrival of type II in their respective regions[15,16,54]. The genetic intermixing of *T. gondii* across nations and continents is believed to be due to human movement and trade[9,55]. Trade between Eastern and Western Asia has ancient roots, with East and West Asia engaging in active trade through the Silk Road by around the 2nd century BCE[56]. Furthermore, from the 1st century A.D. onwards, maritime routes along the coasts were established, connecting China with diverse ports in Southeast Asia, India, Sri Lanka, the Arabian Peninsula, and the Horn of Africa. These maritime routes are suggested to have been the pathways through which HG13 and Africa 4 (#20) spread to coastal countries[16].

The present study does not address the topic of "how each ancestral lineage born". However, by inferring migration and admixture through the recombination history of the type II, HG13, and Africa 4 (#20) ancestral clades, we try to develop a hypothesis about the long-term migration history of clade D *T. gondii*. (1) Initially, the HG2 lineage migrated from the Middle East via the Silk Road in the 2nd century BCE, where it met and interbred with the ancestor of HG13 (Fig. 7). (2) This was followed by interbreeding with Africa 4, which migrated via the Maritime Silk Road. The exact timing of the interaction between the two lineages is unknown, and it is possible that the order of transmission was the opposite. Repeated backcrossing between the hybrids and the parental species, resulting in the introgression of the HG2 and Africa 4 genome into the monochromatic ancestral population of HG13. (3) The related lineage migrated to eastern Siberia and then to North America via the Bering Strait. Land animals were only able to migrate across the Bering Strait during the glacial epochs, when substantial ice sheets established a land bridge between the two continents. However, the last land bridge formed more than 10,000 years ago, and the timing of the land bridge's existence may differ from the

timing of the emergence of the ancestral lineage of HG12 (a HG2–HG13 admixture). This suggests that the migration of ancestral HG12 to North America might not have occurred via the Bering land bridge, and that humans, migratory birds, or marine mammals may have been responsible for this migration. While HG12 is an admixed strain of HG2 and HG13, it shows almost no evidence of admixture with the Africa 4 lineage. It should be noted that this does not indicate that the divergence from HG2–HG13 to HG12 occurred specifically between points (1) and (2); it only broadly suggests the process of HG12's formation. Future isolation and genomic analysis of *T. gondii* on either the Eurasian or Alaskan side near the Bering Strait may further clarify the pathway. (4) This lineage is thought to have partially incorporated the HG11 genome, consequently evolving into HG12. The fact that large haploblocks of the Africa 4 (#20) have not mixed with lineages other than the current HG13 and some Japanese strains may be attributed to the later-arising HG2–HG13-Africa 4 admixed strain (yellow, red, blue), which swept the original HG13 lineage in East Asia. There is a possibility of discovering an ancient type of HG13 by examining *T. gondii* from wild cats in China that have not yet been isolated. (5) Archeological investigations have discovered fossil of domestic cats at ancient Japanese heritage (Karakami site in Iki island, Supplementary Fig. S1), with accelerator mass spectrometry carbon-14 dating indicating these cats date back to the 2nd century BCE[57]. This is considered to mark the introduction of the domestic cat into Japan, suggesting that the *T. gondii* population arriving with these cats was admixed HG2.

The TgCatJp5 strain, characterized by its cyan color in Okinawa, is considered an ancestral lineage before it became fragmented due to frequent recombination events in South/Central America. An ancestral presence of a *T. gondii* population somewhere is likely associated with the historical presence of ancient cats or wild felines in that area. In Japan, two subspecies of the leopard cat are found: the Iriomote wild cat (*Prionailurus bengalensis iriomotensis*), exclusive to Iriomote Island in Okinawa, and the Tsushima wild cat (*Prionailurus bengalensis euptilurus*). The divergence between the Iriomote and Tsushima wild cats dates back to approximately 200,000 years ago[58]. During this

period, the Ryukyu Islands (Nansei-shoto) and the Asian continent were intermittently connected by land bridges, a fact that aligns with the DNA data and geological formation of the Okinawa Islands. Additionally, the leopard cat is present in Taiwan, which is geographically close to Iriomote Island. These hypotheses can be further tested in the future by including more Asian isolates in the analysis, particularly from Southeast Asia.

We recognize several biases should be considered as limitations of this study. One primary challenge is the representativeness of our sampling. Currently, genomic data from West Asia, Central Asia, Southeast Asia, and Russia especially, the Far-East Asian region are absent, leaving a significant gap in covering the genetic diversity present across the vast Asian continent. This gap limits our ability to determine the continuity of certain haplogroups (HG2–HG13, HG13–HG12) across different regions. Therefore, collecting samples from these areas is essential for more comprehensive analysis. Furthermore, the dataset employed in the present study was insufficient for the fine-scale genetic distances or specific timing of divergence between intra-lineage strains. Future analysis with a larger number of isolates within the same group will provide a better understanding of the genetic microstructure. Our present study lays the groundwork for such comprehensive future research. Additionally, the samples from Western Japan used in this study were limited to only two strains isolated in in the Shikoku region, excluding Okinawa. Therefore, the Jp5[cyan] lineage may also exist in the Kyusyu region, which is geographically close to Okinawa. Furthermore, the distance between the Okinawa Islands and Taiwan or Shanghai is similar to that of the Kyusyu region in Japan, suggesting the possibility that this lineage could have been introduced via Southeast Asia or China. It would be helpful to study how common the Jp5[cyan] lineage of *T. gondii* is in Southeast Asia. This will aid in understanding the spread of the parasite. Our present study is a first step in adding information about the *T. gondii* genome from East Asia to the global knowledge base.

Furthermore, using ME49-based probes poses a significant challenge in accurately capturing *T. gondii* strains from different HGs in the sample. This could potentially bias the sequencing data towards specific sequences and hinder our understanding of the diversity of *T. gondii*. The hybrid capture is not affected by a single base substitution in the human genome[59]. Additionally, it has been reported that various strains of SARS-CoV-2 can be detected using probes originally designed based on the Wuhan strain genome sequence[60]. Further validation is necessary to determine the detection of different HGs. However, given that haploblocks significantly different from HG2 were detected in TgCatJp1 and TgCatJp2, it has not occurred that genomes other than HG2 are entirely excluded.

Previous studies classified TgCtCo5, TgRsCr01, and TgCkCr10 as belonging to HG15. However, the neighbor-net analysis conducted in this study showed that TgCkCr10 did not cluster with the other two strains. Additionally, the POPSICLE plot revealed that the genomic structure of TgCtCo5 and TgRsCr01 is a complex admixture including a small portion of HG15[orange] and undetermined clades. Since this HG is probably composed of diverse recombinant genomes, more isolates are needed to infer the ancestral lineage of this population. While POPSICLE is a tool for interpreting admixture, it may not provide a perfect model. Thus, it's important to interpret the results in correlating with other analytical methods and previous knowledge.

In summary, we have conducted the first genome-wide population genomics research involving *T. gondii* strains derived from Far East Asia. Technically, we have demonstrated the power of POPSICLE to deduce ancestral lineages and the utility of Target Enrichment sequencing to resurrect genomes from non-cultivatable isolates. The analysis of new *T. gondii* isolates on an island in the Far East Asia by POPSICLE has provided us with previously unknown ancestral lineages in North and South America and a link between *T. gondii* in Far East Asia and the Americas, conceptually advancing our understanding of the

molecular epidemiology and extent to which recombination has impacted the population genetics and transmission of *T. gondii*. Further hunting new *T. gondii* strains all over the world using genome-wide SNP information in combination with the POPSICLE analysis should provide new insight into the global population structure and pathogenesis of this organism in the future.

## Methods

### Ethic statement
All animal experiments were performed with the approval of the Animal Research Committee of the Research Institute for Microbial Diseases, Osaka University (permission number: R03-20-0).

### Isolation and collection of *T. gondii* Strains
Among the Japanese *T. gondii* strains used in this study, TgCatJpTy1/k3, TgCatJpGi1/TaJ, TgCatJpObi1, and TgCatJp5 were obtained from the laboratory where they were initially isolated. TgBoJp1 and TgBoJp2 were isolated in this study, and the details can be found in the supplementary information. The genomes of ME49, GT1, and CTG were newly sequenced in this study. Other strains were obtained by downloading whole-genome sequence data from previous studies[4,15,23,61]. The complete list of strains used in this study was shown in Supplementary Table S2.

### Bioassay in mice
TgBoJp1 and TgBoJp2 were isolated in this study, originating from wild boars in Ehime, located in central Japan. Wild boars were hunted, and their hearts were transported to Osaka University within 3 days. The cardiac apex (approximately 10-20 g) was added to 150 ml of tissue lysate, mixed until liquid using a mixer, and then dissolved with Proteinase K at 56 °C overnight. The next morning, DNA was extracted using Qiagen's DNeasy Blood & Tissue kit and nested-PCR targeting the B1 gene of *T. gondii* was performed[62]. The oligonucleotides corresponding to the B1 gene were as follows: B1-T1: AGCGTCTCTCTT CAAGCAGCGTA, B1-T2: TCCGCAGCGACTTCTATCTCTCTGT, B1-T3: TGGGAATGAAAGAGACGCTAATGTG, B1-T4: TTAAAGCGTTCGTGGT CAACTATCG. For PCR-positive hearts, approximately 20 g stored at 4 °C were homogenized in 400 ml of 0.9% NaCl containing 0.4% trypsin and 40 µg/ml gentamicin. The mixture was incubated at 37 °C for 90 min[9]. The digests were subjected to two repeated processes using three layers of two-woven gauze to remove the debris, and the filtered solution was centrifuged at 2000 × *g*, 4 °C, for 10 min. The process of washing the pellet with Saline (0.9% NaCl) was repeated twice. The washed pellet was dissolved in 9 ml of Saline (containing antibiotics), and 1 ml was intra-peritoneally inoculated into a total of 9 mice (five *Ifngr1*[−/−] mice and four ICR mice). Peritoneal exudate cells from the affected mice were collected and inoculated into Vero cells (CCL-81, ATCC), to isolate tachyzoites.

### Target enrichment with SureSelect
The samples consisted of a total of 7 strains (1 from Hokkaido, 6 from Okinawa) isolated in Japan between 2011 and 2013 (Supplementary Table S2). Six strains from Okinawa were grown in mice, infected with MDCK cells and frozen. Additionally, 1 strain from Hokkaido was infected with tachyzoites in HFF cells (NB1RGB, RIKEN BRC CELL BANK) and frozen. While mixed infections are suspected when targeted enrichment strategies are used for tissue DNA from wild animal or human samples, but this is unlikely in this study because the samples were isolated by cultured cells before freezing. These vials were stored for approximately 10 years, then recently packaged on dry ice, and shipped by air. Although they were lost as isolates due to failure to revive, DNA was extracted from the small pellets that were removed upon thawing using the QIAGEN's DNeasy Blood & Tissue kit. Based on the threshold cycle of real-time PCR, the amount of *T. gondii* DNA in the specimen was found to be 0.0009% to 0.015% of host DNA. Real-

time PCR utilized Power SYBR Green Master Mix, and the primer sequences used were B1Fw: GAAAGCCATGAGGCACTCCA and B1Rv: TTCACCCGGGACCGTTTAGC. For designing 120mer RNA bait sequences, the *T. gondii* ME49 genome was assembled from recent long read sequencing (in preparation for publication). Briefly, a probe library comprising 263,719 probes covering 49.4% of the entire reference genome (64 Mb) was created by tiling 120 bp long baits without gaps, with one bait acquired by every other bait for a genome size of approximately 64 Mb. After optimizing the probe presence ratios based on GC rates, the final probe library consisted of 360,164 probes. These probes were manufactured by Agilent Technologies (Santa Clara, USA). Library preparation was carried out with the SureSelect HS XT2 kit according to the manual, and multiplex pooled libraries adjusted to 0.5 ng/μl were sent to the Genome Analysis Laboratory, Center for Genetic Information Experimentation, Osaka University. Whole-genome resequencing was performed on a DNBSEQ machine manufactured by MGI Tech (Shenzhen, China).

## Whole genome sequencing

The *T. gondii* strains were cultured in Vero monolayers treated with RPMI (supplemented with 10% fetal bovine serum (FBS), penicillin/streptomycin, and 2-ME). Tachyzoites were collected from Vero cells and filtered through a 5.0 μm filter to eliminate any host cell contamination. DNA from the filtered parasites was extracted using Qiagen's DNeasy Blood & Tissue kit. The extracted DNA from the *T. gondii* strains was sent to the Genome Analysis Laboratory, at the Center for Genetic Information Experimentation, Osaka University, for total genome resequencing using Illumina's Nova-Seq sequencing platform. The fastq accession numbers corresponding to the data used in this study can be found in Supplementary Table 2.

## Variant calling and phylogenetic analysis

All strains were aligned to the ME49 reference genome assembly (version 57) obtained from ToxoDB Version 8.2 (http://ToxoDB.org) using the custom script, including BWA 0.7.17 for read alignment and GATK 4.2.6.1 for read quality control, filtering, and genomic alignment[63]. A customized GATK pipeline was employed for read alignment and SNP calling. High-quality variants were obtained by genotyping SNP loci across the whole genome and excluding low-quality variants, resulting in a set of 1,436,410 high-quality loci in 57 isolates (DP > 3, MQ > 20, QD > 2.0). These SNP lists were then used for downstream analysis. The VCF files containing SNP information were converted into SNP matrices in NEXUS and PHYLIP formats using the Python script "vcf2phylip v2.6"[64]. The NEXUS files were directly incorporated into SplitsTree v4.18.2 (http://www.splitstree.org) and a rootless phylogenetic tree network was constructed using the neighbor-net method. The network was depicted with 1000 bootstrap support greater than 95%[65]. Subchromosome networks were also created by extracting regions of interest from chromosomes using GATK-SelectVariant from a file containing SNP variant information for all chromosomes. The apicoplast genome referenced ToxoDB-61-*T. gondii*RH88, and a TCS network was created. For the analysis, only SNPs mapped by three or more reads were used, and five strains (FOU, GUY-KOE, MAS, P89, RUB) showing more than 5% missing sites were excluded from the analysis.

## SNP density plots

The SNP density analysis was performed as previously described[66]. Initially, GATK and VCFtools were used to convert the VCF file covering the entire genome into a tabular format that exclusively included biallelic SNPs without INDELs. Positions of SNPs that differed from the reference sequence were extracted, resulting in the inclusion of only non-reference sequence positions for each strain's data. The count of SNPs within a specified window of 10 kbp in length was calculated for each strain. Consequently, higher number of SNPs within a given

window indicated a greater deviation from the reference genome (ME49).

## Assessment of in vivo virulence in mice

Eight-week-old female C57BL/6J mice and ICR mice were obtained from Japan SLC (Shizuoka, Japan). The mice were intraperitoneally infected with 1000 tachyzoites of the RH, ME49, and TgCatJp5 strains and their survival was monitored for up to 30 days. Additionally, *Ifngr1*$^{-/-}$ mice previously generated and described were used[67]. All experiments were conducted using 8–10-week-old mice.

## Immunofluorescence assay

MEF cells were seeded in 12-well plates and treated with IFN-γ (10 ng/ml) before being infected with the parasite for 2 h (infection multiplicity = 4). Following infection, cells were fixed in 4% paraformaldehyde in PBS for 10 min at room temperature, permeabilized with 0.2% digitonin, and blocked with 8% bovine serum albumin in PBS. Coverslips were then incubated with the primary antibody for 1 h at room temperature followed by incubation with a fluorescent secondary antibody for 1 h at room temperature. Goat polyclonal antibody against IRGB6 (1:100) was purchased from Santa Cruz Biotechnology, Inc (sc-11079). Rabbit polyclonal anti-GBP2 (H00008878, Proteintech, USA). Mouse monoclonal anti-GRA2 (1:100) antibody was provided by D. Soldati-Favre (University of Geneva). A rabbit polyclonal anti-GRA7 antibody (1:100) was provided by John C. Boothroyd (Stanford University). Nuclei were counterstained with Hoechst dye. The coverslips were mounted on glass slides using PermaFluor (Thermo Scientific) and images were captured using a confocal laser microscope (Olympus FV3000 IX83). The accumulation of IRG molecules in PVs was determined by randomly selecting cells for analysis.

## Copy number variation analysis

Sequencing depth was calculated at every 100 bp using tinycov (v.0.4.0, https://github.com/cmdoret/tinycov). Copy number variation was determined by dividing the maximum sequencing depth of the ROP5 gene region (566,721–568,370 bp) by the average sequencing depth of chromosome XII. The mapping status of ROP18 was calculated as the relative sequencing depth of the gene region containing the UPS region (chromosome VIIa: 1,516,000–1,517,000 bp).

## Population genomic structure

Population genomic structure was assessed using POPSICLE (https://POPSICLE-admixture.sourceforge.io by Shaik, JS., Khan, A. & Grigg, ME.) with a sliding window size of 10 kb. The samples were compared to the reference sequence ME49 with a range of cluster numbers (K = 2–15). The optimal number of clusters was determined using the Dunn index[36]. Once the optimal number of clusters was obtained, POPSICLE assigned each block to an existing or new clade based on the population structure of the sample and the ancestral status of each block. A Circos plot[68] was then generated, with colors assigned based on the cluster number. Chromosome paintings were created using POPSICLE Admixture (https://sourceforge.net/projects/POPSICLE-admixture/files/data/). Further cluster analysis for benchmarking was performed by ADMIXTURE v.1.3[35] and fastSTRUCTURE[34]. The VCF files were converted to PED format using PLINK v.1.9[69] as input, imported into ADMIXTURE and fastSTRUCTURE, and plotted using R scripts available at https://fujinitaka.hatenablog.com/entry/2020/01/03/005453. We downloaded fastq files using the SRA toolkit for the data published in the previous study[4] and performed variant calling. The fastq file of GUY-2004-ABE that we downloaded was corrupted for some reason, so we excluded this strain and performed the analysis on 61 strains.

## Reporting summary

Further information on research design is available in the Nature Portfolio Reporting Summary linked to this article.

## Data availability

Source data are provided as a Source Data file. The sequence data generated in this study have been deposited in the DDBJ database under accession code DRR513065, DRR513066, DRR513067, DRR513068, DRR513069, DRR513070, DRR513071, DRR513072, DRR513073, DRR513074, DRR513075, DRR513076, DRR513077, DRR513078, DRR513079, DRR513080, DRR513081, The sequence data published in previous studies were used under accession code. SRX160127, SRX156300, SRX099787, SRX099792, SRX159844, SRX159890, SRX057823, SRX038728, SRX038699, SRX160141, SRX099794, SRX055419, SRX055414, SRX099773, SRX055412, SRX099796, SRX099783, SRX099783, SRX038725, SRX046278, SRX099774, SRX160123, SRX099788, SRX160124, SRX099804, SRX099805, SRX099795, SRX160134, SRX099791, SRX038693, SRX055420, SRX038727, SRX099779, SRX160142, SRX055413, SRX038726, SRX055418, SRX055416, SRX099782, SRX171132, SRX099803, SRX099776, SRR366806, SRR350724, SRX099793, SRX099774, SRX156168, SRX156037, SRX155963, SRX155534, SRX160125, SRX159841, SRX156192, SRX156164, SRX156155, SRX154747, SRX099784, SRX160143, SRX099789, SRX099775, SRX160050, ERS13421665, ERS13421666, ERS13421667, Nucleus genome referenced ToxoDB-57-T.gondiiME49 [https://toxodb.org/common/downloads/release-57/TgondiiME49/fasta/data/ToxoDB-57_TgondiiME49_Genome.fasta], Apicoplast genome referenced ToxoDB-61-T.gondiiRH88 [https://toxodb.org/common/downloads/release-61/TgondiiRH88/fasta/data/ToxoDB-61_TgondiiRH88_Genome.fasta] Source data are provided with this paper.

## Code availability

The codes for the analyses presented in this paper, including the BWA-GATK and POPSICLE, are available in GitHub (https://github.com/Fumiakii2/Supplemental-code, DOI: 10.5281/zenodo.10785333). Other codes are described in detail in Methods and available in published papers and public websites or, for in-house pipelines, is available upon reasonable request from the corresponding author.

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

## Acknowledgements

We thank M. Enomoto and N. Yamagishi (Osaka University) for secretarial and technical assistance. We also thank Dr. J.C. Boothroyd and Dr. D. Soldati-Favre for anti-GRA7 antibody and anti-GRA2 antibody, respectively. We thank Dr. S. Jilong and Dr. T. Irie for *T. gondii* DNA of WH3 strain and *Sarcocystis spp.* DNA, respectively. This study was supported by Japan Science and Technology Agency (JPMJFR206D and JPMJMS2025); Agency for Medical Research and Development (JP20fk0108137, JP23fk0108682, and JP223fa627002); Ministry of Education, Culture, Sports, Science and Technology (20B304 and 19H00970); Japan Society for the Promotion of Science (22K20614); the program from Joint Usage and Joint Research Programs of the Institute of Advanced Medical Sciences, Tokushima University; Takeda Science Foundation; Mochida Memorial Foundation; Astellas Foundation for Research on Metabolic Disorders; Naito Foundation; the Chemo-Sero-Therapeutic Research Institute; Research Foundation for Microbial Diseases of Osaka University; BIKEN Taniguchi Scholarship; The Nippon Foundation - Osaka University Project for Infectious Disease Prevention; Joint Research Program of Research Center for Global and Local Infectious Diseases of Oita University (2021B06); and in part by the Intramural Research Program of the National Institute of Allergy and Infectious Diseases (NIAID) at the National Institutes of Health.

## Author contributions

F.I., M.E.G. and M.Y. designed research; F.I. performed research; F.I., H.K., Y.T., T.M., M.I., Y.N., K.Hikosaka and H.S. provided parasites strains; F.I., F.O., K.Hayashi., K.Y., M.S., S.F., M.S., H.I., K.K., Y.O. and A.Y. isolated parasite strains; F.I., S.N. and D. M. sequenced *T. gondii* genome; F.I. and M.Y. wrote the first draft paper; F.I., M.S., M.E.G. and M.Y. wrote and edited the paper.

## Competing interests

The authors declare no competing interests.

## Additional information

[1]Department of Immunoparasitology, Research Institute for Microbial Diseases, Osaka University, Suita, Osaka 565-0871, Japan. [2]Laboratory of Immunoparasitology, WPI Immunology Frontier Research Center, Osaka University, Suita, Osaka 565-0871, Japan. [3]Department of Immunoparasitology, Center for Infectious Disease Education and Research, Osaka University, Suita, Osaka 565-0871, Japan. [4]Okinawa Prefectural Institute of Health and Environment, Uruma, Okinawa 904-2241, Japan. [5]Faculty of Applied Biological Sciences, Gifu University, Gifu 501-1112, Japan. [6]Center for One Medicine Translational Research, COMIT, Gifu University, Gifu 501-1112, Japan. [7]Department of Veterinary Associated Science, Faculty of Veterinary Medicine, Okayama University of Science, Imabari, Ehime 794-8555, Japan. [8]Laboratory of Parasitology, Faculty of Veterinary Medicine, Okayama University of Science, Imabari, Ehime 794-8555, Japan. [9]Joint Faculty of Veterinary Medicine Kagoshima University, Kagoshima 890-0065, Japan. [10]National Research Center for Protozoan Diseases, University of Agriculture and Veterinary Medicine, Obihiro, Hokkaido 080-8555, Japan. [11]Department of Infection and Host Defense, Graduate School of Medicine, Chiba University, Chiba 260-0856, Japan. [12]Department of Infection Metagenomics, Research Institute for Microbial Diseases, Osaka University, Suita, Osaka 565-0871, Japan. [13]Laboratory of Wildlife Management, Faculty of Agriculture, Iwate University, Morioka, Iwate 020-8550, Japan. [14]Laboratory of Veterinary Parasitology, Faculty of Agriculture, Iwate University, Morioka, Iwate 020-8550, Japan. [15]Laboratory of Veterinary Anatomy, University of Agriculture and Veterinary Medicine, Obihiro, Hokkaido 080-8555, Japan. [16]Laboratory of Veterinary Parasitology, School of Veterinary Medicine, Kitasato University, Aomori 034-8628, Japan. [17]Division of Risk Analysis and Management, International Institute for Zoonosis Control, Hokkaido University, Sapporo, Hokkaido 001-0020, Japan. [18]Laboratory of Veterinary Parasitic Diseases, Department of Veterinary Sciences, Faculty of Agriculture, University of Miyazaki, Miyazaki 889-2155, Japan. [19]Center for Animal Disease Control, University of Miyazaki, Miyazaki 889-2155, Japan. [20]Molecular Parasitology Section, Laboratory of Parasitic Diseases, National Institutes of Health, National Institute of Allergy and Infectious Diseases (NIAID), Bethesda, MD 20892, USA. ✉e-mail: myamamoto@biken.osaka-u.ac.jp

