## [Peer Review File · Nature Communications]

Far-East Asian Toxoplasma isolates share ancestry with North and South/Central American recombinant lineagesREVIEWER COMMENTS

Reviewer #1 (Remarks to the Author):

In this study, the authors have explored the ancestry patterns of Far-East Asian populations of *Toxoplasma gondii*. Appropriate tools were used for ancestry analyses. The analysis focused on a number of Japanese *T. gondii* strains. This work provides evidence for shared ancestry between Japanese strains and other Asian strains, and also between Japanese strains and certain North and South American *T. gondii* populations. This work confirms the results of previous studies that have shown that Asian and North American *T. gondii* populations are related (Lorenzi et al., 2016; Galal et al., 2022). It shows for the first time that links between Asian and South American populations also exist.

Lorenzi, et al. "Local admixture of amplified and diversified secreted pathogenesis determinants shapes mosaic *Toxoplasma gondii* genomes." *Nature communications* 7.1 (2016): 10147.

Galal, et al. "A unique *Toxoplasma gondii* haplotype accompanied the global expansion of cats." *Nature Communications* 13.1 (2022): 5778.

Beyond these aspects, the analysis falls short for several reasons:

- The dataset is far from being representative of Far-East Asian countries (~20 countries) since only China and Japan are represented. Moreover, the analysis is only based on a small number of Japanese and Chinese *T. gondii* strains.
- No fine scale / within lineage analyses were conducted in order to provide elements on the place of Far-East Asian *T. gondii* in their respective groups / lineages. This could have helped in evaluating genetic distances between strains of the same groups (e.g., type II) but of distinct geographical origins and therefore to provide clues on their timing of divergence.
- No inference on the evolutionary history of Far-East Asian *T. gondii* populations is proposed in relation to the history of exchanges between these regions and in link with hosts dissemination.
- A major *T. gondii* lineage in Asia (Africa4/#20) has been omitted from the analyses although it is available on public databases.

A detailed phenotypic analysis of a virulent strain was included in the manuscript. The reviewer considers that this part is a digression from the main topic of the study, and that this part affects the overall balance of the manuscript.

Secondary remarks:

L85: This is a confusion between two notions. A lineage is a biological reality, in the sense that it is a strain that only multiplies asexually. Haplogroups are lineages or strains that cluster together in a distance tree and this a much less constituent concept, since haplogroups will probably change according to the strains included in the tree. For example, type II belongs to HG2 but is not HG2. This confusion should be resolved throughout the manuscript.

L88: please change "large" for "wild" throughout the manuscript.

L89: This is not exactly the case. Common clonal lineages (Africa1/BrI/#6 in Brazil, type II in Chile) exist beside recombinant strains that have more or less spread clonally.

L92: Africa4 and #20 are the same: It is at least found in Egypt, Tunisia, Senegal, Benin, Arab Peninsula, Sri Lanka and China.

Dong, et al. "Isolation, genotyping and pathogenicity of a *Toxoplasma gondii* strain isolated from a Serval (*Leptailurus serval*) in China." *Transboundary and emerging diseases* 66.4 (2019): 1796-1802.

L122: If the authors refer to the #4 genotype, this genotype is mainly found in domestic animals (Jiang et al., 2018). #5 is the wild one.

Jiang, et al. "A partition of *Toxoplasma gondii* genotypes across spatial gradients and among host

species, and decreased parasite diversity towards areas of human settlement in North America." *International Journal for Parasitology* 48.8 (2018): 611-619.

L180: "When SNPs were restricted to high confidence" This formulation is unclear.

L186: It is more a distance analysis than a phylogenetic analysis.

L192: The reviewer does not see any blocks supporting recombination for these two strains. The same very short regions referred to as blocks are also found in PRU, a type II strain as ME49.

L312: This work is not providing evidence on recombinations.

L316: The reviewer does not understand how the authors are inferring on kinship directionality here. In addition, other scenarios have been proposed elsewhere (Galal et al., 2022). Galal, et al. "A unique *Toxoplasma gondii* haplotype accompanied the global expansion of cats." *Nature Communications* 13.1 (2022): 5778.

L395: Is this formulation not overinterpreting the directionality of this divergence?

L398: The sister group of a given group is the one closest to it in a phylogenetic tree. HG12 is closer to HG2 than to HG11.

Fig.2 C-D: These figures are not very clear. The coordinates (check the spelling) are not very clear. It is hard to understand where are the boundaries of the genes and how to deduce the information on genes copy numbers from these graphs.

L422: An ancestral presence of a *T. gondii* population somewhere should be associated to the presence of an ancient presence of cats or of wild felids in this area. Is it the case for this island?

L428: The North American wild genotype #5 has also been reported in China

L655-670: The accession numbers of fastq files and of the project should be provided rather than biosamples.

Reviewer #2 (Remarks to the Author):

In this work, Ihara et al present a study of the population structure of the apicomplexan parasite *Toxoplasma gondii* in Japan based on whole genome sequencing data generated from 17 parasite strains isolated from Japan, seven of them using a novel target-enrichment approach, and two additional strains from China. By applying several known bioinformatic tools commonly used for population structure studies, the authors present evidence of the presence of two novel ancestral parasite lineages, one shared with haplogroup (HG) 12 parasites from North America and the second lineage, which seems to be confined to strains from the Japanese island of Okinawa, also found in strains from clade F and HG15 from Central and northern South America. Whole-genome-based *T. gondii* population studies from this part of the world has been lacking and this work fills this gap in knowledge and provides new insights into the transmission and evolution of this important parasite. Therefore, the results and findings presented herein will be of interest for the research communities working on parasite ecology and evolution and on public health issues associated with Toxoplasmosis in particular and with other apicomplexan parasites. Despite its novelty and importance, I found that the current work presents several potential methodological issues and concerns that the authors will need to address before I can recommend its publication:

Major revisions:

1- The authors applied a novel SureSelect Target Enrichment System using probes based on the ME49 genome sequence to enrich infected mammalian cell samples for *T. gondii* DNA before

sequencing. How this technique deals with potential mix-infections with strains from different HGs? Also, it is likely that *T. gondii* genomic regions highly divergent from ME49 are negatively selected and therefore not represented in the sequencing data. The authors should discuss this potential limitation of the methodology used and reinterpret their results, if needed.

2- Lines 168-185: Ihara et al propose that TgCatJpOk4 represents a major *T. gondii* group in Okinawa, given that appears "close" to rescued strain TgCatJp1 in the neighbor-network tree of Figure S3A. However, they also state that TgCatJpOk4 appears "far" from other strains from clade D in the same Figure when, at the naked eye, TgCatJpOk4 seems to be located midway from TgCatJp1 and ARI. Similarly, the separation between TgCatJp1 and TgCatJpOk4 strains appears to be relatively similar to the separation of TgCatJp1 and the root from HG3 strains. The authors should provide additional evidence supporting the hypothesis that TgCatJpOk4 represents a major group in Okinawa. For example, do you see combination of haploblocks that are both conserved between TgCatJpOk4 and TgCatJp1 and that include some blocks from the Jp5-cyan ancestor and/or JpOk4-magenta? Or what you were actually trying to convey here was that crosses between Japanese HG2 and TgCatJp5-related strains seem to have happened several times in the past, as discussed in Lines 447-449? If so, please make it clearer.

3- Lines 230-232 and Lines 599-604 (methods): The estimation of ROP5 low copy number in TgCatJp5 seems to be incorrect. The *T. gondii* ME49 genome version used for the analysis contains 2 copies of the ROP5 gene, which explains the double peak of approximate the same height in Figure 2C. Therefore, the copy number should have been estimated by multiplying the mean max sequencing depth of both peaks multiplied by 2 and divided by the mean read depth of chromosome XII, which for TgCatJp5 gives an estimated copy number of ~9 copies of ROP5. This leads to my second question: which ROP5 copy from each genome was chosen for building the phylogenetic trees of Figure 2E? Based on the methods section (lines 590-597), the sequence used was derived from the identified SNPs in ROP5 with GATK's FastaAlternateReferenceMaker, which replaces specific bases from a reference sequence with the corresponding SNPs from a VCF file. This causes several issues. First, in the VCF file, ROP5 bases that are highly polymorphic across copies from the same strain could have potentially been filtered out because they did not pass the SNP filtering cutoffs. Similarly, underrepresented variants (e.g. SNPs that are present in only one ROP5 instance but not in the rest of the genes in the array) could have been missed. Second, in those instances where all variants from one position are correctly annotated in the VCF file, FastaAlternateReferenceMaker will just pick one at random to be incorporated in the final fasta sequence. Third, assuming that the region chosen for replacing SNPs spanned only one of the two annotated ROP5 copies from ME49 v57 (XII:566,721-568,370 as suggested by the methods), all the SNPs called on the second copy would have been missed. Finally, because VCF files do not keep track of SNPs that are phased, the resulting consensus sequence used for the analysis was likely a chimeric sequence from all the ROP5 copies present in the strain. Considering all these potential issues, my suggestion is to use a different approach to generate as accurately as possible the sequences of the different copies of ROP5 present in each strain before running the analysis of Figure 2E. Otherwise, the tree from Figure 2E could be completely misleading.

4- Lines 288-290 and Fig. 3A: using the proposed $k = 9$ for the POPSICLE analysis the clustering algorithm separates strain TgCtCo5 from TgRsCr01 and TgCkCr10, when the three are classified as HG15 and the haplogroup proportions seem to be closer between TgCtCo5 and TgRsCr01 (central circle). Is this an artifact of the k value selected or what is the explanation? Please discuss.

5- The authors suggest that "highly pathogenic hybrids could emerge in Okinawa" similar to what has been described with South American *T. gondii* strains. This is based on (i) virulence assessment of TgCatJpOk4 and TgCatJp5 in animal models, (ii) a reported higher seropositivity rate in the island compared to the rest of Japan, and (iii) the cases of porcine toxoplasmosis reported in the period 2018-2022 in Japan where almost 100% of them were from Okinawa. However, higher seropositivity rates and number of symptomatic swine could be also explained by socio-economic and cultural differences between Okinawa and the rest of Japan. Please provide references supporting the stats about porcine toxoplasmosis and discuss potential alternative explanations.

6- In the methods section, the authors provide a brief description of the software used for this work. However, I find it difficult to assess the quality of the SNP data generated, and eventually reproduce the results, without having access to the actual code showing programs, versions and parameters used, which can nowadays be easily shared via GitHub or any other software repository. Please include links to this information, including pipeline for generating SNP data and commands used for running POPSICLE, unless the authors used all default parameters as

described in the POPSICLE documentation. Also, provide as supplementary data the DNA sequences of the ROP5 and ROP18 genes used for the phylogenetic analyses of Figure 2E. In addition, on lines 544 and 545 Ihara et al indicate that variant calling was performed "... using the recommended best practices, ...". Please provide a reference to the source of the followed best practices (e.g. ToxoDB, GATK?).

Minor revisions:

- 1- Throughout the text: "Latin America". Latin America does not have a precise definition and the term does not include South American countries Guyana and Suriname. Please use a more precise definition (e.g. South America, Central America).
- 2- Line 87: "example. haplogroup". please remove "."
- 3- Fig 1C: Strain names are incorrectly aligned.
- 4- Please define clearly in the text the meaning of the term "sequencing coverage" you use in the manuscript. From the text, it is not clear if you are referring to depth of sequencing, which seems to be the case, or the span of the sequencing data across the *T. gondii* genome.
- 5- Line 229: Fix typo "ROP1".
- 6- Lines 196-197: Please add TgCatJpGi1/Taj to the list of TgBoJp1 and TjBoJp2, since it does not share the end of chr Ia with the other sequenced Japanese strains.
- 7- Line 600: Add reference/link to tinycov program.
- 8- Line 250: "...using sophisticated programs...". Please remove "sophisticated" from this sentence, given that is just a personal, subjective opinion.
- 9- Lines 447-449: "The presence of TgCatJpOk1". Did you mean TgCatJp1?
- 10- Figure 1C: Strain labels are not aligned correctly with the heatmap. Please adjust.

Reviewer #3 (Remarks to the Author):

I think this has potential to serve as an important contribution. The authors use innovative laboratory techniques (probe capture hybridization) and bioinformatics (Popsicle) and include samples from a heretofore poorly sampled region (East Asia) to add to our understanding of genetic structure for *Toxoplasma gondii*.

As expressed below, my greatest concerns are:

1. Even after their work, East Asia remains poorly sampled. I fear the manuscript does not fully acknowledge the limitations that still surround this matter, or explain to readers the statistical strength of inferences they draw.
2. They present a somewhat anecdotal approach to accepting or rejecting analytic conclusions.
3. They generate interesting new hypotheses (a connection between isolates in Okinawa, Japan and certain isolates from South America), seek evidence to confirm this possibility- and arguably make a lot out of little evidence. I believe these stand as interesting hypotheses to explore with future data.
4. They deploy an over broad concept of Latin America. While it is true that Tropical South America (Amazonian Brazil, French Guiana) harbor populations of *Toxoplasma* notably lacking in clonal strains, other parts of Latin America (temperate portions of Brazil, say, and portions of the Caribbean) do have clonal lineages and important constituents.

60, 88, 416- it might be accurate to say that no clonal structure exists in "portions of Latin America" or, more precisely, in tropical Amazonia. But this is not true for Latin America as a whole,

where temperate regions (even in Brazil) do harbor clonal strains.

97 "It has been shown.....are thought to be" rephrase: Recently derived recombinant strains contribute.....

104 defend "a small number of ancestral lineages" or omit
114 "is known to be HG1 is known" rephrase

123 , based on the alleles found..... sequences, (add commas)

126 , isolated from a US AIDS patient. (add comma)

142 including isolates from China

168 What do you mean by "the most abundant Japanese strains?" Do you have prevalence data? Or do you just mean 3 of haplotypes?

250 readers do not benefit from your assessment that these programs are sophisticated and widely used, but they could benefit from a succinct description of what they do/how they work (as they did for Popsicle on line 268).

264 This conclusion concerns me. Who decides which results deserve to be considered "correct" or "reasonable." The authors depict very flat curves for these tools, indicating that the data contain too little signal, or too much conflict, to confidently decide how many ancestral populations may have given rise to the extant data. They also show that the two tools disagree on this point. The truth may lie somewhere between, or near, 6-8.

Is this a problem with the tools? Or a reflection of uncertainty in the data? I urge the authors to consider that these models, though powerful, cannot perfectly mimic the underlying population and genetic processes.

279 Perhaps say that these results better conform to prior assumptions. Whether they "demonstrate more validity" seems to presume that they already know the answer...which seems dubious.

Also, if they embrace 14 subdivisions, I think it would be interesting to go back to fastStructure and ADMIXTURE and run those models under that assumption. Then, do these tools converge on a shared pattern of subdivision among the isolates? Figure S6 depicts using K=14 for ADMIXTURE and fastSTRUCTURE. These results deserve discussion, given the preference of the authors for that assumption.

282 This logic confuses me. Explain how including other, less related populations, would undermine the goal? By placing all of the Japanese isolates into one or just a few groups?

287 Again I wonder about the criteria....who decides when subdivision is "unnecessary" and on what grounds? Aesthetic preference? Going back to the previous comment, if you added less closely related lineages, does the tool now group these three HG15 lineages?

293 HG3, a recombinant line,

377 Do you take this as the basis of a conclusion, or the basis of establishing a hypothesis? I enjoyed this "part of the journey" but I am a little uncomfortable about the seemingly post hoc nature of seeking confirmation for a particular connection here.

383 fairer to say that Asia was underrepresented. I would not agree that all regions except Asia have enjoyed careful attention. Much of Africa, for example, remains poorly characterized.

385-8 The addition of these several isolates from Japan and China are very welcome. But I think, in fairness, the authors should acknowledge this is a good start. But Asia is vast, and a mere handful of isolates will likely bias inferences of population diversity and structure.

421 while plausible, this idea really needs tempering by acknowledging the very small sample here. Okinawa may have maintained an isolated lineage. But where have we still not sampled? Will our view change?

One technique the authors have not used is to resample data, with replacement. How stable is their view to stochastic variation in the present dataset? Would removing some data or some isolates reshape the view, or are the highlighted findings robust to such perturbations? The Haplotype network (554) was constructed with bootstrapping as was the RAXML analysis of ROP5

and ROP18 (597). What about other results.

436 Yes I like this approach.... Here you illustrate what questions future data can help resolve. I think this paper would benefit by adopting this perspective more consistently.

454 remove "the" in front of 165 cases

First of all, we greatly appreciate that the reviewers have dedicated to providing insightful feedback on ways to strengthen our paper. We are therefore pleased to resubmit our article for further consideration. We have incorporated changes that reflect the detailed suggestions you have graciously provided. We hope our revisions and the responses adherently address the concerns by the reviewers.

Please find below our point-by-point response to your comments and suggestions.

(Comments 1-7 are correspondence to the editor, therefore Comment 8 is the first.)

Point-by-point responses to Reviewer #1:

Comment 8: In this study, the authors have explored the ancestry patterns of Far-East Asian populations of Toxoplasma gondii. Appropriate tools were used for ancestry analyses. The analysis focused on a number of Japanese T. gondii strain. This work provides evidence for shared ancestry between Japanese strains and other Asian strains, and also between Japanese strains and certain North and South American T. gondii populations. This work confirms the results of previous studies that have shown that Asian and North American T. gondii populations are related (Lorenzi et al., 2016; Galal et al., 2022). It shows for the first time that links between Asian and South American populations also exist.

AUTHORS' RESPONSE: We extend our sincere thanks to the reviewer for his/her insightful comments. These comments have been invaluable in helping us enhance the quality and clarity of our paper.

*Comment 9: Beyond these aspects, the analysis falls short for several reasons:
- The dataset is far from being representative of Far-East Asian countries (~20 countries) since only China and Japan are represented. Moreover, the analysis is only based on a small number of Japanese and Chinese T. gondii strains.*

AUTHORS' RESPONSE: We wish to thank the reviewer for this comment. Indeed, as noted, 'Far East Asia' often encompasses countries in Southeast Asia. However, in our study, the term 'Far East Asia' specifically refers to Japan, China, South Korea, Mongolia, and Taiwan, and excludes Southeast Asian countries such as Indonesia, Malaysia, and the Philippines. To preclude any potential misunderstanding, we have now included a clarifying explanation in our text. Additionally, in response to another valuable comment, we have incorporated three African4(#20) strains into our analysis.

Accordingly, we have changed this text to:

(p. 6, line 131- 134):

“Far East Asia’ often encompasses countries in Southeast Asia. However, in our study, the term 'Far East Asia' specifically refers to Japan, China, South Korea, Mongolia, and

Taiwan, and excludes Southeast Asian countries such as Indonesia, Malaysia, and the Philippines.”

Comment 10:- No fine scale / within lineage analyses were conducted in order to provide elements on the place of Far-East Asian T. gondii in their respective groups / lineages. This could have helped in evaluating genetic distances between strains of the same groups (e.g., type II) but of distinct geographical origins and therefore to provide clues on their timing of divergence.

AUTHORS’ RESPONSE: We are grateful for the insightful scientific comment provided. We acknowledge the importance of a detailed analysis of genetic diversity within lineages to better understand the history of divergence within the same group. In response, we have conducted an additional Neighbor-net analysis, that includes type II, type X, and HG2-like Japanese strains (see **Fig. S3C**).

Accordingly, we have added:.

(p. 8-9, line 182-185):

"An intra-group analysis of type II, HG12, and HG2-like Japanese strains showed that three strains—TgCatJpOk3, TgCatJpTy1/k3, and TgCatJpObi1—were distinct from both type II and HG12 strains (**Fig. S3C**). Among them, TgCatJpObi1 specifically exhibited evidence of recombination with HG12."

(p. 23, line 589-593):

“Furthermore, the dataset employed in the present study was insufficient for the fine-scale genetic distances or specific timing of divergence between intra-lineage strains. Future analysis with a larger number of isolates within the same group will provide a better understanding of the genetic microstructure. This study lays the groundwork for such comprehensive future research.”

Comment 11: No inference on the evolutionary history of Far-East Asian T. gondii populations is proposed in relation to the history of exchanges between these regions and in link with hosts dissemination.

AUTHORS' RESPONSE: We deeply appreciate the reviewer's suggestion regarding the inclusion of an evolutionary perspective in our study. Recognizing its importance for understanding the distribution and genetic diversity of this pathogen, we have now incorporated hypotheses related to evolutionary aspects into our discussion section. Furthermore, to gain insights into the evolutionary history, we have added an analysis of the maternally inherited apicoplast genome. This analysis, based on sequences referenced in ToxoDB61_RH, involved creating a TCS network. We used only SNPs mapped by three or more reads for this analysis, excluding five strains (FOU, GUY-KOE, MAS, P89, RUB) due to more than 5% missing data. The findings, particularly the relationship between Africa 4 (#20) and HG13, have yielded intriguing insights into the regional exchange of *T. gondii*. In addition, we have included a graphical summary illustrating the journey of Clade D *Toxoplasma gondii*, considering recombination.

Accordingly, we have changed this text to:

Apicoplast Method

(p. 28-29, line 722-725):

“The apicoplast genome referenced ToxoDB-61-*T. gondii*RH88, and a TCS network was created. For the analysis, only SNPs mapped by three or more reads were used, and five strains (FOU, GUY-KOE, MAS, P89, RUB) showing more than 5% missing sites were excluded from the analysis.”

Apicoplast Result

(p. 9, line 190-200):

“Africa 4 (#20) cluster was located at the boundary of clade D. HG13 appears genetic recombination of HG2 and Africa 4 (#20).

Additionally, an analysis of the apicoplast genome (an organelle inherited maternally) identified 177 SNPs in the 35 Kbp genome. Most Japanese strains inherited the apicoplast from HG2 (**Fig. 1B**). HG13 haplotype was linked to TgCatJp5 and HG12 (RAY and B41) by a few SNPs, suggesting their kinships. ARI shared its genome with COUG (HG11) strain. The genetic distance between HG13 and HG2 suggests that they diverged from a common ancestor and then followed different evolutionary pathways.

Africa 4 (#20) lineage was distinct from other haplotypes, suggesting unique genetic characteristics.”

Evolutionary hypothesis

(p. 21-23, line 533-568):

“The type II, estimated to have appeared about 26,000 years ago¹⁶, was selected for by the domestication of the Libyan wildcat in the Fertile Crescent with the advent of agriculture about 10,000 years ago, and subsequently spread to Europe and Asia⁵⁵. It is known that agriculture also began in China and parts of Africa, suggesting that the ancestral lineages of Africa 4 (#20) and HG13 predated the arrival of type II in their respective regions^{16,56,57}. The genetic intermixing of *T. gondii* across nations and continents is believed to be due to human movement and trade^{10,58}. Trade between Eastern and Western Asia has ancient roots, with East and West Asia engaging in active trade through the Silk Road by around the 2nd BCE⁵⁹. Furthermore, from the 1st century AD onwards, maritime routes along the coasts were established, connecting China with diverse ports in Southeast Asia, India, Sri Lanka, the Arabian Peninsula, and the Horn of Africa. These maritime routes are suggested to have been the pathways through which HG13 and Africa 4 (#20) spread to coastal countries⁵⁷.

The present study does not address the topic of "how each ancestral lineage born". However, by inferring migration and admixture through the recombination history of the type II, HG13, and Africa 4 (#20) ancestral clades, we try to develop a hypothesis about the long-term migration history of clade D *T. gondii*.

- ① Initially, the HG2 lineage migrated from the Middle East via the Silk Road in the 2nd century BC, where it met and interbred with the ancestor of HG13 (**Fig. 7, red arrow**).
- ② This was followed by interbreeding with Africa 4, which migrated via the Maritime Silk Road. The exact timing of the interaction between the two lineages is unknown, and it is possible that the order of transmission was the opposite (**Fig.7, blue arrow**). Repeated backcrossing between the hybrids and the parental species, resulting in the introgression of the HG2 and Africa 4 genome into the monochromatic ancestral population of HG13.
- ③ The related lineage migrated to eastern Siberia and then to North America via the Bering Strait.
- ④ This lineage is thought to have partially incorporated the HG11 genome, consequently evolving into HG12. The fact that large

haploblocks of the Africa 4 (#20) have not mixed with lineages other than the current HG13 and some Japanese strains may be attributed to the later-arising HG2-HG13-Africa 4 admixed strain (yellow, red, blue), which swept the original HG13 lineage in East Asia. There is a possibility of discovering an ancient type of HG13 by examining *T. gondii* from wild cats in China that have not yet been isolated. ⑤ Archaeological investigations have discovered fossil of domestic cats at ancient Japanese heritage (Karakami site in Iki island, **Fig. S1**), with AMS carbon-14 dating indicating these cats date back to the 2nd century BC⁶⁰. This is considered to mark the introduction of the domestic cat into Japan, suggesting that the *T. gondii* population arriving with these cats was admixed HG2.”

(p. 44, line 1136-1141):

Figure legend:

“Figure 7. Journey of Clade D *Toxoplasma* Considering Recombination.

A hypothetical diagram illustrating the historical migration events of Clade D *Toxoplasma*. The sequence of events from 1 to 5 corresponds to specific genetic recombination episodes. The red arrows represent the directions of genetic admixture. Asterisks on HG13 and HG12 indicate that these genetic lineages had ancestral-type genomes, which differed from their current genome.”

Comment 12: A major T. gondii lineage in Asia (Africa4/#20) has been omitted from the analyses although it is available on public databases.

AUTHORS’ RESPONSE: We are grateful for the important point highlighted by the reviewer. We concur with reviewer #2's suggestion regarding the value of Africa4/#20 strains in understanding the African-Asian *T. gondii* connection. Accordingly, in this revised version, we have included the available three Africa4(#20) strains from Senegal.

(p. 8, line 158-161):

“This dataset included 2 or 3 representative strains from the 16 haplogroups plus 3 Africa 4(#20) strains along with 8 Japanese strains, including 6 newly sequenced strains (**Fig. S1 shows the regions where they were isolated**).”

(p. 17-18, line 424-443):

“Evaluation of the genetic distance among the HG2, HG13, and Africa 4 genomes.

To characterize the Africa 4^{blue} genomic region, which constitutes approximately 30% of the HG13 genome, we focused on the Africa 4^{blue} region in both TgCatPRC2 (HG13) and 160504Fcat01 (Africa 4) (**Fig. 6A**). A neighbor-net analysis showed that the Africa 4^{blue} region in HG13 is closely related to the Africa 4 (#20) genome (**Fig. 6B**).

Furthermore, we focused on the genomic region where color hues were assigned as HG2^{yellow} in ME49, HG13^{red} in TgCatPRC2, and Africa4^{blue} in 160504Fcat01.

In these unexchanged regions, the strong recombination relationship previously observed between HG13 and Africa 4 (#20) in the whole genome was resolved (**Fig. 1A and Fig. 6C**). In this region (17.04 Mbp), we identified 128,689 SNPs in 160504Fcat01 (Africa 4) compared to 82,500 SNPs in TgCatPRC2 (HG13) (**Table S3**). This is consistent with the apicoplast genome, suggesting a more recent divergence between HG2^{yellow} and HG13^{red} than with Africa 4^{blue}. To investigate the extent to which the Africa 4^{blue} genome is conserved in lineages other than HG13, we examined the proportion of haploblocks within each 10kbp-sliding window. The admixed Japanese HG2 strains had 1.9-2.1%, while only very small fragments (0.1-0.3%) were observed in HG11, HG12, and HG15 (**Table S4**). Altogether, these results suggest that Africa 4 genome contributes to the *T. gondii* populations of HG13 and Japan, as well as to the Indian Ocean coastal region where Africa 4(#20) lineage is frequently distributed (**Fig 6D**).”

Comment 13: A detailed phenotypic analysis of a virulent strain was included in the manuscript. The reviewer considers that this part is a digression from the main topic of the study, and that this part affects the overall balance of the manuscript.

AUTHORS’ RESPONSE: We are grateful for the scientific comment. In response to the reviewer's remarks, and following careful consultation with the editor, we have decided to retain the detailed phenotypic analysis of the virulent strain in our manuscript. We acknowledge the reviewer's concern about potential digression from the main focus of the study and its impact on the manuscript's balance. However, we believe that this detailed phenotypic information provides critical insights that complement our main findings and enhance the overall understanding of the topic.

Secondary remarks:

Comment 14: L85: This is a confusion between two notions. A lineage is a biological reality, in the sense that it is a strain that only multiplies asexually. Haplogroups are lineages or strains that cluster together in a distance tree and this a much less constituent concept, since haplogroups will probably change according to the strains included in the tree. For example, type II belongs to HG2 but is not HG2. This confusion should be resolved throughout the manuscript.

AUTHORS' RESPONSE: We are grateful for the reviewer's comment. In response, we have revised related statements in the manuscript. For example,

(p. 5, line 85-86):

“The *T. gondii* population genomic structure is comprised of 6 major clades that are subdivided into 16 major haplogroups that cluster together in a distance tree⁴.”

(p. 5, line 86-90):

“The three archetypal lineages now belong to Haplogroups (HGs) 1, 2, and 3. Geographic segregation of the other 13 haplogroups is common, with for example. Type X belongs to HG12 mostly isolated from North America, whereas HG11 is largely restricted to wild felids⁵⁻⁷.”

(p. 5, line 93-96):

“Archetypal lineages II and III are also the main lineages in Africa, Africa 1 circulates widely in the West and Central areas, belonging to HG6⁹.” Africa 3 that might have emerged after two or more recombination events between Africa 1, type II, and type III strains, belonging to HG14, are found in Gabon^{10,11}.”

Comment 15: L88: please change “large” for “wild” throughout the manuscript.

AUTHORS' RESPONSE: In line with the reviewer's suggestion, we have replaced the term 'large' with 'wild' in the relevant sections of our manuscript."

(p. 5, line 87-90):

“Geographic segregation of the other 13 haplogroups is common, with for example. Type X belongs to HG12 mostly isolated from North America, whereas HG11 is largely restricted to wild felids⁵⁻⁷.”

(p. 5, line 101-103):

“Recently derived recombinant strains significantly contribute to the current population of isolated *T. gondii*, aside from those strains that originate from distant locales and wild felines¹⁵.”

Comment 16: L89: This is not exactly the case. Common clonal lineages (Africa1/BrI/#6 in Brazil, type II in Chile) exist beside recombinant strains that have more or less spread clonally.

AUTHORS' RESPONSE: We thank the reviewer for this comment. The reviewer's comment is correct. To clarify, we have added the following text to the Introduction.

(p. 5, line 90-93):

“The genetic diversity of *T. gondii* in South/Central America is quite high and differs markedly from Europe and North America(20). Although common clonal lineages (Africa 1 in Brazil, and type II in Chile) exist, highly genetically diverse recombinant strains are expanding⁸.”

Comment 17: L92: Africa4 and #20 are the same: It is at least found in Egypt, Tunisia, Senegal, Benin, Arab Peninsula, Sri Lanka and China.

Dong, et al. "Isolation, genotyping and pathogenicity of a Toxoplasma gondii strain isolated from a Serval (Leptailurus serval) in China." Transboundary and emerging diseases 66.4 (2019): 1796-1802.

AUTHORS' RESPONSE: We appreciate the reviewer's comment on this point. The reviewer's comment is correct. To clarify, we have added the following text to the Introduction.

(p. 5, line 96-99):

“Another genotype Africa 4, referred to as, Restriction Fragment Length Polymorphism (RFLP) #20 has been found in Egypt, Tunisia, Senegal and Benin^{12,13}. Strains with this genotype were also isolated in Sri Lank, Arab Peninsula, and China¹⁴.”

Comment 18: L122: If the authors refer to the #4 genotype, this genotype is mainly found in domestic animals (Jiang et al., 2018). #5 is the wild one.

Jiang, et al. "A partition of Toxoplasma gondii genotypes across spatial gradients and among host species, and decreased parasite diversity towards areas of human settlement in North America." International Journal for Parasitology 48.8 (2018): 611-619.

AUTHORS' RESPONSE: We thank the reviewer for this comment. We corrected the mistake.

(p. 6, line 122-124):

“Interestingly, two strains (TgCatJpObi1 and TgMonkeyJp1) which were isolated in Hokkaido, the north island of Japan (**Fig. S1**), possessed the same RFLP genotype as RFLP#4 mainly found in domestic animals in North America^{22,23}.”

Comment 19: L180: "When SNPs were restricted to high confidence" This formulation is unclear.

AUTHORS' RESPONSE: We thank the reviewer for this comment. We revised the expression to:

(p. 9, line 204-206):

“When restricted to SNPs at least four or more reads mapped for the other 4 strains, the Neighbor network revealed that TgCatJp4, TgCatJp7, and TgMonkeyJp1 belong to the HG2 subgroup (**Fig. S4A, S4C and S4E**),”

Comment 20: L186: It is more a distance analysis than a phylogenetic analysis.

AUTHORS' RESPONSE: We thank the reviewer for this comment. We revised the sentence to:

(p. 10, line 211-212):

“The genome-wide distance analysis indicates a significant amount of admixture in the Japanese strains (**Fig. 1A**).”

Comment 21: L192: The reviewer does not see any blocks supporting recombination for these two strains. The same very short regions referred to as blocks are also found in PRU, a type II strain as ME49.

AUTHORS' RESPONSE: We appreciate the reviewer's comment. We revised the manuscript to:

(p. 10, line 216-218):

“As expected, the two strains (TgBoJp1 and TgBoJp2), which were part of HG2 (**Fig. 1A**), share a few very short blocks that are also found in PRU and a type II strain ME49 (**Fig. 1C**).”

Comment 22: L312: This work is not providing evidence on recombinations.

AUTHORS’ RESPONSE: We appreciate the reviewer’s comment. We revised the manuscript to:

(p. 14, line 346-349):

Furthermore, we found that the POPSICLE analysis classified the HG13 strains (TgCatPRC2 and WH3) as a mosaic including 35% HG2^{yellow} (**Fig. 3D**), which was consistent with a previous study that HG13 shares genomic regions with ME49 (~40%)⁴.

(p. 14, line 353-355):

Moreover, approximately 34% of the HG13 genome was introgressed with the Africa4^{blue}, while only less than 0.1% of the HG12 genome showed such admixture.

Comment 23: L316: The reviewer does not understand how the authors are inferring on kinship directionality here. In addition, other scenarios have been proposed elsewhere (Galal et al., 2022).

Galal, et al. "A unique Toxoplasma gondii haplotype accompanied the global expansion of cats." Nature Communications 13.1 (2022): 5778.

AUTHORS’ RESPONSE: We thank the reviewer for this interesting question. We added a section how we are inferring on kinship directionality in **Comment 11**.

Comment 24: L395: Is this formulation not overinterpreting the directionality of this

divergence?

AUTHORS' RESPONSE: We regret this oversight. To clarify, we have added the following text to:

(p. 15, line 374-376):

Surprisingly, we observed that this magenta hue shared a common root with HG11 (COUG, GUY-JAG1, TgPumaME1) in clade D but that it had accumulated a significant number of SNPs (**Fig. 4C**).

(p. 19, line 460-466):

Additionally, this study identified JpOk4^{magenta}, a genomic region on chromosome V (1.28 Mbp to 2.4 Mbp), located near the genome of HG11 (HG11^{gray}). This region overlaps with a segment (1.25 Mbp to 2.1 Mbp) previously proposed to be similar between HG11 and HG12 in the haplotype map by Minot et al.¹⁵. Given that the region is closer to HG11 than its sister lineages HG2, these results indicate that JpOk4^{magenta} is the region separating and characterizing HG12 from other HGs.

Comment 25: L398: The sister group of a given group is the one closest to it in a phylogenetic tree. HG12 is closer to HG2 than to HG11.

AUTHORS' RESPONSE: We thank the reviewer's comment. We revised the manuscript is in **Comment 24**.

Comment 26: Fig.2 C-D: These figures are not very clear. The coordinates (check the spelling) are not very clear. It is hard to understand where are the boundaries of the genes and how to deduce the information on genes copy numbers from these graphs.

AUTHORS' RESPONSE: We thank the reviewer's comment. We updated the **Fig. 2C and 2D** with diagrams of the locus of ROP5 (_308090) and ROP18(_205250) , and locus surrounding ROP5 (_308075, _308093,a and 308096).

Comment 27: L422: An ancestral presence of a T. gondii population somewhere should be associated to the presence of an ancient presence of cats or of wild felids in this area. Is it the case for this island?

AUTHORS' RESPONSE: We thank the reviewer for this comment. The answer is yes, Iriomote wild cat (*Prionailurus bengalensis iriomotensis*) is a subspecies of the leopard cat found exclusively on the small island of Iriomote in the Okinawa Prefecture. This rare and elusive feline is of significant conservation concern, classified as Critically Endangered on the IUCN Red List.

We revised the manuscript to:

(p. 23, line 569-581):

“The TgCatJp5 strain, characterized by its cyan color in Okinawa, is considered an ancestral lineage before it became fragmented due to frequent recombination events in South/Central America. An ancestral presence of a *T. gondii* population somewhere is likely associated with the historical presence of ancient cats or wild felines in that area. In Japan, two subspecies of the Leopard cat are found: the Iriomote wild cat (*Prionailurus bengalensis iriomotensis*), exclusive to Iriomote Island in Okinawa, and the Tsushima wild cat (*Prionailurus bengalensis euptilurus*). The divergence between the Iriomote and Tsushima wild cats dates back to approximately 200,000 years ago⁶¹. During this period, the Ryukyu Islands (Nansei-shoto) and the Asian continent were intermittently connected by land bridges, a fact that aligns with the DNA data and geological formation of the Okinawa Islands. Additionally, the leopard cat is present in Taiwan, which is geographically close to Iriomote Island. These hypotheses can be further tested in the future by including more Asian isolates in the analysis, particularly from Southeast Asia.”

Comment 28: L428: The North American wild genotype #5 has also been reported in China

AUTHORS' RESPONSE: We thank the reviewer's comment. We revised the manuscript to:

(p. 20, line 497-500):

“Furthermore, HG13, which is predominant in China, has been sporadically identified in the North, Central, and South America (one strain from Colombia, one from Mexico, and one from the United States) and Southeast Asia (one from Sri Lanka and four from Vietnam). The North American wild genotype RFLP#5 has also been reported in China.”

Comment 29: L655-670: The accession numbers of fastq files and of the project should be provided rather than biosamples.

AUTHORS' RESPONSE: We thank the reviewer's comment. We revised the accession numbers to:

ME49_DRR513065; CTG_DRR513066; GT1_DRR513067; TgCatJp5_DRR513068;
TgCatJpTy1/k3_DRR513069; TgCatJpGi1/TaJ_DRR513070;
TgCatJpObi1_DRR513071; TgBoJp1_DRR513072; TgBoJp2_DRR513073;
TgCatJp1_DRR513074; TgCatJp2_DRR513075
TgCatJp3_DRR513076; TgCatJp4_DRR513077; TgCatJp6_DRR513078;
TgCatJp7_DRR513079; TgMonkeyJp1_DRR513080; WH3_DRR513081

Point-by-point responses to Reviewer #2:

Comment 30: In this work, Ihara et al present a study of the population structure of the apicomplexan parasite Toxoplasma gondii in Japan based on whole genome sequencing data generated from 17 parasite strains isolated from Japan, seven of them using a novel target-enrichment approach, and two additional strains from China. By applying several known bioinformatic tools commonly used for population structure studies, the authors present evidence of the presence of two novel ancestral parasite lineages, one shared with haplogroup (HG) 12 parasites from North America and the second lineage, which seems to be confined to strains from the Japanese island of Okinawa, also found in strains from clade F and HG15 from Central and northern South America. Whole-genome-based T. gondii population studies from this part of the world has been lacking and this work fills this gap in knowledge and provides new insights into the transmission and evolution of this important parasite. Therefore, the results and findings presented herein will be of interest for the research communities working on parasite ecology and evolution and on public health issues associated with Toxoplasmosis in particular and with other apicomplexan parasites. Despite its novelty and importance, I found that the current work presents several potential methodological issues and concerns that the authors will need to address before I can recommend its publication:

AUTHORS' RESPONSE: We extend our sincere appreciation to the reviewer for their insightful comments on our paper. These observations have been invaluable in helping us enhance the quality and clarity of our work."

Major revisions:

Comment 31: 1- The authors applied a novel SureSelect Target Enrichment System using probes based on the ME49 genome sequence to enrich infected mammalian cell samples for T. gondii DNA before sequencing. How this technique deals with potential mix-infections with strains from different HGs? Also, it is likely that T. gondii genomic regions highly divergent from ME49 are negatively selected and therefore not represented in the sequencing data. The authors should discuss this potential limitation of the methodology used and reinterpret their results, if needed.

AUTHORS' RESPONSE: We appreciate the reviewer's insightful comment on this matter. We concur that using ME49-based probes poses a significant challenge in accurately capturing *T. gondii* strains from different haplogroups (HGs) in the sample, which could potentially introduce a bias in the sequencing data towards certain sequences, thus limiting our understanding of *T. gondii* diversity. In response to this issue, we have created coverage plots for all SureSelect sequencing data (**Fig. S2**). Additionally, we have acknowledged this as one of the limitations of our study in the revised manuscript.

Furthermore, we agree with the reviewer's comment regarding mixed infections. However, this particular concern does not apply to the samples used in our study, as the samples were isolated by limiting dilution in cultured cells before being frozen. When using target enrichment strategy for tissue DNA from animals or human samples, the challenge is to distinguish mixed infections of different clones. We have also added this point to Method. In case where mixed infections are suspected, for example, when using target enrichment strategy for tissue DNA from wild animals or human samples, we may be able to convert these heterozygous SNP loci to homozygous by 'phasing' to haploid haplotypes across the genome (ALFRED AMAMBUA-NGWA, et al. 2019, Science)

(p. 8, line 171-175):

“Coverage plots were created for seven samples, revealing varying depth of coverage (**Fig. S2**). The reads were mostly uniformly distributed across the entire chromosome, with a few regions showing significantly low depth. These regions corresponded to low-complexity sequences in the reference genome. “

(p. 16, line 409-412):

“Furthermore, the additional POPSICLE analysis identified haploblocks such as HG1^{navy}, HG3^{brown}, and HG6^{deepgreen}, which deviate from HG2 in TgCatJp1 and TgCatJp2 (**Fig. S10D**). This indicates that hybrid capture can capture sequences from different HGs for ancestor analysis, even with ME49-based probes.”

(p. 24, line 603-612):

“Furthermore, using ME49-based probes poses a significant challenge in accurately capturing *T. gondii* strains from different HGs in the sample. This could potentially bias the sequencing data towards specific sequences and hinder our understanding of the diversity of *T. gondii*. The hybrid capture is not affected by a single base substitution in the human genome⁶². Additionally, it has been reported that various strains of SARS-CoV-2 can be detected using probes originally designed based on the Wuhan strain genome sequence⁶³. Further validation is necessary to determine the detection of different HGs. However, given that haploblocks significantly different from HG2 were detected in TgCatJp1 and TgCatJp2, it has not occurred that genomes other than HG2 are entirely excluded.”

(p. 27, line 672-675):

“While mixed infections are suspected when targeted enrichment strategies are used for tissue DNA from wild animal or human samples, but this is unlikely in this study because the samples were isolated by cultured cells before freezing.”

Comment 32: 2- Lines 168-185: Ihara et al propose that TgCatJpOk4 represents a major T. gondii group in Okinawa, given that appears “close” to rescued strain TgCatJp1 in the neighbor-network tree of Figure S3A. However, they also state that TgCatJpOk4 appears “far” from other strains from clade D in the same Figure when, at the naked eye, TgCatJpOk4 seems to be located midway from TgCatJp1 and ARI. Similarly, the separation between TgCatJp1 and TgCatJpOk4 strains appears to be relatively similar to the separation of TgCatJp1 and the root from HG3 strains. The authors should provide additional evidence supporting the hypothesis that TgCatJpOk4 represents a major group in Okinawa. For example, do you see combination of haploblocks that are both conserved between TgCatJpOk4 and TgCatJp1 and that include some blocks from the Jp5-cyan ancestor and/or JpOk4-magenta? Or what you were actually trying to convey here was that crosses between Japanese HG2 and TgCatJp5-related strains seem to have happened several times in the past, as discussed in Lines 447-449? If so, please make it clearer.

AUTHORS' RESPONSE: We are deeply grateful for the reviewer's insightful comment. We concur that our original explanation of the neighbor-net analysis displayed a bias influenced by the findings from POPSICLE on genomic structure. Accordingly, we have revised our interpretation to be a more analytical. In response to the reviewer's observation, we acknowledge the limitations of this network in predicting the haplotypes of TgCatJpOk4 and TgCatJp1. As a result, we have excluded these strains from the major groups identified in Okinawa.

Furthermore, to explore the possibility of shared haploblocks between TgCatJpOk4 and TgCatJp1, we newly generated a POPSICLE plot that includes TgCatJp1 and TgCatJp2. This analysis revealed that the TgCatJp1 genome serves as an intermediate between TgCatJp5 and TgCatJpOk4, suggesting a kinship with TgCatJpOk4.

Accordingly, we have changed this text to:

(p. 9, line 187-188):

“Furthermore, TgCatJpOk4 appeared to be located midway between the branches of HG12 and HG13, suggesting that it is a recombinant strain.”

(p. 9, line 207-210):

“TgCatJp1 was located between TgCatJpOk4 and HG3, appearing to be a distinct recombinant (**Fig.4A**). These results suggest that TgCatJp5 is a major group on the southern islands of Japan. “

(p. 16, line 405-408):

“Additionally, to explore the possibility of shared haploblocks between TgCatJpOk4 and TgCatJp1, we generated a POPSICLE plot that includes TgCatJp1 (**Fig. S10D**). This revealed that the TgCatJp1 genome serves as an intermediate between TgCatJp5 and TgCatJpOk4, suggesting a kinship with TgCatJpOk4. “

Comment 33: 3- Lines 230-232 and Lines 599-604 (methods): The estimation of ROP5 low copy number in TgCatJp5 seems to be incorrect. The T. gondii ME49

genome version used for the analysis contains 2 copies of the ROP5 gene, which explains the double peak of approximate the same height in Figure 2C. Therefore, the copy number should have been estimated by multiplying the mean max sequencing depth of both peaks multiplied by 2 and divided by the mean read depth of chromosome XII, which for TgCatJp5 gives an estimated copy number of ~9 copies of ROP5. This leads to my second question: which ROP5 copy from each genome was chosen for building the phylogenetic trees of Figure 2E? Based on the methods section (lines 590-597), the sequence used was derived from the identified SNPs in ROP5 with GATK's FastaAlternateReferenceMaker, which replaces specific bases from a reference sequence with the corresponding SNPs from a VCF file. This causes several issues. First, in the VCF file, ROP5 bases that are highly polymorphic across copies from the same strain could have potentially been filtered out because they did not pass the SNP filtering cutoffs. Similarly, underrepresented variants (e.g. SNPs that are present in only one ROP5 instance but not in the rest of the genes in the array) could have been missed. Second, in those instances where all variants from one position are correctly annotated in the VCF file, FastaAlternateReferenceMaker will just pick one at random to be incorporated in the final fasta sequence. Third, assuming that the region chosen for replacing SNPs spanned only one of the two annotated ROP5 copies from ME49 v57 (XII:566,721-568,370 as suggested by the methods), all the SNPs called on the second copy would have been missed. Finally, because VCF files do not keep track of SNPs that are phased, the resulting consensus sequence used for the analysis was likely a chimeric sequence from all the ROP5 copies present in the strain. Considering all these potential issues, my suggestion is to use a different approach to generate as accurately as possible the sequences of the different copies of ROP5 present in each strain before running the analysis of Figure 2E. Otherwise, the tree from Figure 2E could be completely misleading.

AUTHORS' RESPONSE: We are thankful for the precise and insightful comment provided by the reviewer. We fully agree with the expressed concerns regarding our ROP5 analyses.

At first, the reviewer #2 said;

"3- Lines 230-232 and Lines 599-604 (methods): The estimation of ROP5 low copy number in TgCatJp5 seems to be incorrect. The T. gondii ME49 genome version used for the analysis contains 2 copies of the ROP5 gene, which explains the double peak of

approximate the same height in Figure 2C. Therefore, the copy number should have been estimated by multiplying the mean max sequencing depth of both peaks multiplied by 2 and divided by the mean read depth of chromosome XII, which for TgCatJp5 gives an estimated copy number of ~9 copies of ROP5. “

Update of Fig. 2C: In response to the comment from the reviewer#2 regarding ROP5 CNV analysis. In light of this, we have reanalyzed the data using the method described by Behke et al. 2015 (PLOS Genetics) resulting in updated versions of Fig. 2C and 2D. These revisions aim to clarify and correct the interpretation of the data. To ensure reproducibility of the results, we added a ROP5 CNV of archetypal type I, II, and III in **Fig. S7B**.

In addition, the reviewer #2 also said;

“This leads to my second question: which ROP5 copy from each genome was chosen for building the phylogenetic trees of Figure 2E? Based on the methods section (lines 590-597), the sequence used was derived from the identified SNPs in ROP5 with GATK’s FastaAlternateReferenceMaker, which replaces specific bases from a reference sequence with the corresponding SNPs from a VCF file. This causes several issues. First, in the VCF file, ROP5 bases that are highly polymorphic across copies from the same strain could have potentially been filtered out because they did not pass the SNP filtering cutoffs. Similarly, underrepresented variants (e.g. SNPs that are present in only one ROP5 instance but not in the rest of the genes in the array) could have been missed. Second, in those instances where all variants from one position are correctly annotated in the VCF file, FastaAlternateReferenceMaker will just pick one at random to be incorporated in the final fasta sequence. Third, assuming that the region chosen for replacing SNPs spanned only one of the two annotated ROP5 copies from ME49 v57 (XII:566,721-568,370 as suggested by the methods), all the SNPs called on the second copy would have been missed. Finally, because VCF files do not keep track of SNPs that are phased, the resulting consensus sequence used for the analysis was likely a chimeric sequence from all the ROP5 copies present in the strain. Considering all these potential issues, my suggestion is to use a different approach to generate as accurately as possible the sequences of the different copies of ROP5 present in each strain before running the analysis of Figure 2E. Otherwise, the tree from Figure 2E could be completely misleading.”

We are considering the three following responses.

As the reviewer #2 pointed out, the tree of ROP5 in Fig. 2E may be misleading. To address potential misinterpretation, we considered reanalyzing this part as following. First, we will obtain sequences of ROP5 alleles as defined in previous studies (Behke et al., 2011, PNAS, Niedelman et al. 2012, Plos Pathogen). Second, we will map raw reads against the different ROP5 alleles. Third, generate ROP5 alleles for each strain by GATK-independent method such as Pilon program. Then we will perform phylogenetic analysis. However, we must acknowledge that even with these adjustments, fully addressing reviewer #2's concerns may be challenging. This is due to the high similarity between different ROP5 alleles, such as ROP5A, ROP5B, and ROP5C, which complicates the accurate phylogenetic analysis of ROP5 to the extent desired by the reviewer.

So, we just deleted Fig. 2E. Then, we updated **Fig.2C and D** to compensate for the loss of the data in **Fig. 2E**, we included TgCatJpOk3 in the CNV analysis of ROP5 and ROP18. If **Fig. 2E** and the corresponding sentences are removed from RESULTS, the hypothesis for the inheritance of virulence factors among the Japanese three strains described in **Fig. 2F** (it is **Fig. 2E** now) can be suggested by the data of Fig. 2C and 2D.

Accordingly, we have changed this text to:

(p. 11, line 254-268):

“Referring to ROP5 copy number of archetypal I, II, and III, TgCatJpOk4 exhibited type III ROP5, as described previously³⁰ (**Fig. 2C and Fig.S7C**). Moreover, TgCatJpOk4 possessed type II ROP18 allele (**Fig. 2D**). We found that TgCatJp5 carries a low copy number, highly active type III ROP5 (ROP5^{III}) allele, and a low activity type III ROP18 (ROP18^{III}) allele with some deletion in the UPS region (**Fig. 2C and D and Fig.S7C**). The Japanese HG2 (TgCatJpOk3) possessed inactive ROP5 (ROP5^{II}) and active ROP18 (ROP18^{II}). Based on the results so far, the low virulent strains were attributed to having inactive ROP5^{II} in the Japanese HG2 and inactive ROP18^{III} in TgCatJp5. However, TgCatJpOk4 had active ROP18^{II} and active ROP5^{III}. Altogether, the likely explanation for the high virulence of TgCatJpOk4 is due to the inheritance of ROP18^{II} from the HG2 (such as TgCatJpOk3) and ROP5^{III} from TgCatJp5 lineage (**Fig. 2E**), as was previously observed for the S23 recombinant F1 progeny that became

mouse-virulent after inheriting the ROP18^{II} allele x ROP5^{III} allele from mouse-avirulent parent strains ME49 (type II) and CTG (type III), respectively³⁵.”

Comment 34: 4- Lines 288-290 and Fig. 3A: using the proposed k = 9 for the POPSICLE analysis the clustering algorithm separates strain TgCtCo5 from TgRsCr01 and TgCkCr10, when the three are classified as HG15 and the haplogroup proportions seem to be closer between TgCtCo5 and TgRsCr01 (central circle). Is this an artifact of the k value selected or what is the explanation? Please discuss.

AUTHORS’ RESPONSE: We appreciate the reviewer’s scientific comment. TgCtCo5 was classified as HG15 in the neighbor-net analysis. Despite its admixture pattern being more similar to TgRsCr01, the inner frame of TgCtCo5 was assigned to HG6 (dark green) in our previous POPSICLE plot below. This classification results from the POPSICLE inner frame mechanically picking colors that represent the clades constituting the strain. In the case of TgCtCo5, it exhibited slightly less orange genome (HG15), compared to the other strains like TgRsCr01 and TgCkCr10, hence dark green was chosen. While POPSICLE is a tool for interpreting admixture, it does not provide a perfect model. Thus, it’s important to interpret its results in correlating with other analytical methods and previous knowledge. Based on the results of the revised POPSICLE plot (**New Fig. 3A**), this limitation is also noted in the Discussion section.

Accordingly, we have changed this text to:

(p. 24, line 613-621):

“Previous studies classified TgCtCo5, TgRsCr01, and TgCkCr10 as belonging to HG15. However, the neighbor-net analysis conducted in this study showed that TgCkCr10 did not cluster with the other two strains. Additionally, the POPSICLE plot revealed that the genomic structure of TgCtCo5 and TgRsCr01 is a complex admixture including a small portion of HG15^{orange} and undetermined clades. Since this HG is probably composed of diverse recombinant genomes, more isolates are needed to infer the ancestral lineage of this population. While POPSICLE is a tool for interpreting admixture, it may not provide a perfect model. Thus, it’s important to interpret the results in correlating with other analytical methods and previous knowledge.”

Comment 35: 5- The authors suggest that “highly pathogenic hybrids could emerge in Okinawa” similar to what has been described with South American T. gondii strains. This is based on (i) virulence assessment of TgCatJpOk4 and TgCatJp5 in animal models, (ii) a reported higher seropositivity rate in the island compared to the rest of Japan, and (iii) the cases of porcine toxoplasmosis reported in the period 2018-2022 in Japan where almost 100% of them were from Okinawa. However, higher seropositivity rates and number of symptomatic swine could be also explained by socio-economic and cultural differences between Okinawa and the rest of Japan. Please provide references supporting the stats about porcine toxoplasmosis and discuss potential alternative explanations.

AUTHORS’ RESPONSE: We are grateful for the reviewer’s comment and agree with this point. The discussion has been revised to incorporate considerations of social and cultural factors that might contribute to the high prevalence of antibody positivity in humans. We have also added citations for porcine toxoplasmosis.

Accordingly, we have changed this text to:

(p. 21, line 519-532):

“There is no significant difference in the seropositivity rate of *T. gondii* in fattening pigs (6-7 months) between Okinawa and mainland Japan^{52,53}. However, out of the 166 cases of symptomatic porcine toxoplasmosis diagnosed during slaughter inspections in Japan over the past five years (2018-2022), 165 cases were identified in Okinawa. Post-slaughter inspections revealed swelling, hemorrhage, and necrosis of internal lymph nodes, as well as hemorrhage of the liver in most affected pigs. Furthermore, although the national average rate of *T. gondii* antibody positivity among Japanese is approximately 6%, which is low compared to other countries’ rates, antibody possession in Okinawa has been found to be as high as 12.9% among humans^{19,54}.

The higher seropositivity rates and number of symptomatic swine could be explained by socio-economic and cultural differences between Okinawa and the rest of Japan. However, highly pathogenic hybrids such as TgCatJpOk4 could also play a significant

role. Consequently, isolating and analyzing *T. gondii* from symptomatic swine in Okinawa is needed to enhance our understanding of the risks to public health.”

Comment 36: 6- In the methods section, the authors provide a brief description of the software used for this work. However, I find it difficult to assess the quality of the SNP data generated, and eventually reproduce the results, without having access to the actual code showing programs, versions and parameters used, which can nowadays be easily shared via GitHub or any other software repository. Please include links to this information, including pipeline for generating SNP data and commands used for running POPSICLE, unless the authors used all default parameters as described in the POPSICLE documentation. Also, provide as supplementary data the DNA sequences of the ROP5 and ROP18 genes used for the phylogenetic analyses of Figure 2E. In addition, on lines 544 and 545 Ihara et al indicate that variant calling was performed “... using the recommended best practices, ...”. Please provide a reference to the source of the followed best practices (e.g. ToxoDB, GATK?).

AUTHORS’ RESPONSE: We appreciate the reviewer’s comment on this point. We have prepared and uploaded a comprehensive 'POPSICLE TUTORIAL' our GitHub repository. Additionally, we have included the script used for BWA-GATK in the same repository, ensuring accessibility and transparency of our methods.
<https://github.com/Fumiakii2/Supplemental-code>

Minor revisions:

Comment 37: 1- Throughout the text: “Latin America”. Latin America does not have a precise definition and the term does not include South American countries Guyana and Suriname. Please use a more precise definition (e.g. South America, Central America).

AUTHORS’ RESPONSE: We thank the reviewer for this comment. Accordingly, we have changed "Latin America" to "South/Central America" throughout the manuscript, including in the title, to ensure accuracy and clarity in our geographical references.

Comment 38: 2- Line 87: “example. haplogroup”. please remove “.”

AUTHORS’ RESPONSE: In accordance with the reviewer's comment, we have changed this to:

(p. 5, line 86-90):

The three archetypal lineages now belong to Haplogroups (HGs) 1, 2, and 3. Geographic segregation of the other 13 haplogroups is common, with for example type X belongs to HG12 mostly isolated from North America, whereas HG11 is largely restricted to wild felids⁵⁻⁷.

Comment 39: 3- Fig 1C: Strain names are incorrectly aligned.

AUTHORS’ RESPONSE: We thank the reviewer for this comment. We corrected strain names. Fig.2C is now **Fig. S6**.

Comment 40: 4- Please define clearly in the text the meaning of the term “sequencing coverage” you use in the manuscript. From the text, it is not clear if you are referring to depth of sequencing, which seems to be the case, or the span of the sequencing data across the T. gondii genome.

AUTHORS’ RESPONSE: We thank for the reviewer’s comment on this point. We were referring to the depth of sequencing. We have changed “sequencing coverage” to “sequencing depth” throughout the manuscript.

Comment 41:5- Line 229: Fix typo “ROPI”.

AUTHORS' RESPONSE: This error has been corrected in accordance with the reviewer's comment.

Comment 42:6- Lines 196-197: Please add TgCatJpGi1/Taj to the list of TgBoJp1 and TgBoJp2, since it does not share the end of chr Ia with the other sequenced Japanese strains.

AUTHORS' RESPONSE: We appreciate the reviewer's comment on this point. In accordance with the reviewer's comment, we have changed this to:

(p. 10, line 221-223):

The block at the end of chromosome Ia was common to all Japanese strains except for TgBoJp1, TgBoJp2, and TgCatJpGi1/TaJ (**Fig. 1C red arrow**).

Comment 43:7- Line 600: Add reference/link to tinycov program.

AUTHORS' RESPONSE: In accordance with the reviewer's comment, we have added link for tinycov.

(p. 30, line 759-760):

Sequencing depth was calculated at every 100 bp using tinycov (<https://github.com/cmdoret/tinycov>).

Comment 44:8- Line 250: "...using sophisticated programs...". Please remove "sophisticated" from this sentence, given that is just a personal, subjective opinion.

AUTHORS' RESPONSE: We appreciate the reviewer's comment on this point. We have deleted the word “sophisticated” from this sentence.

(p. 12, line 272-274):

Therefore, we performed cluster analysis to infer ancestral structure using programs such as ADMIXTURE and fastSTRUCTURE, both of which are used in population genetics^{36,37}.

Comment 45:9- Lines 447-449: “The presence of TgCatJpOk1”. Did you mean TgCatJp1?

AUTHORS' RESPONSE: We thank the reviewer for this comment. The comment is correct, we corrected this typo.

(p. 21, line 514-516):

The presence of TgCatJp1 indicates that the recombinant, like TgCatJpOk4, has been repeatedly produced, suggesting that such hybrid *T. gondii* genomes are endemic in this area.

Comment 46:10- Figure 1C: Strain labels are not aligned correctly with the heatmap. Please adjust.

AUTHORS' RESPONSE: We thank the reviewer for this comment. We corrected strain names. Fig.2C is now **Fig. S6**.

Point-by-point responses to Reviewer #3:

Comment 47:

*I think this has potential to serve as an important contribution. The authors use innovative laboratory techniques (probe capture hybridization) and bioinformatics (Popsicle) and include samples from a heretofore poorly sampled region (East Asia) to add to our understanding of genetic structure for *Toxoplasma gondii*.*

AUTHORS' RESPONSE: We express our sincere gratitude to the reviewer for their insightful comments, which have significantly contributed to enhancing the consistency and clarity of the conclusions derived from our study results.

As expressed below, my greatest concerns are:

Comment 48:

1. Even after their work, East Asia remains poorly sampled. I fear the manuscript does not fully acknowledge the limitations that still surround this matter, or explain to readers the statistical strength of inferences they draw.

AUTHORS' RESPONSE: We recognize several biases should be considered as limitations of this study. One primary challenge is the representativeness of our sampling. Currently, genomic data from West Asia, Central Asia, Southeast Asia, and Russia especially, the Far-East Asian region are absent, leaving a significant gap in covering the genetic diversity present across the vast Asian continent. This gap limits our ability to determine the continuity of certain haplogroups (HG2-HG13, HG13-HG12) across different regions. Therefore, collecting samples from these areas is essential for more comprehensive analysis. Additionally, a better understanding the distribution of genotypes such as TgCatJpOk4^{Magenta} and TgCatJp5^{Cyan}, which so far have been identified only in Japan within Asia, is crucial to grasp the full scope of *T. gondii*'s spread and propagation. This study represents an initial step in integrating the *T. gondii* genome from East Asia into global knowledge and how future research can address these gaps.

Accordingly, we have changed this text to:

(p. 23, line 582- 588):

“We recognize several biases should be considered as limitations of this study. One primary challenge is the representativeness of our sampling. Currently, genomic data from West Asia, Central Asia, Southeast Asia, and Russia are absent, leaving a significant gap in covering the genetic diversity present across the vast Asian continent. This gap limits our ability to determine the continuity of certain haplogroups (HG2-HG13, HG13-HG12) across different regions. Therefore, collecting samples from these areas is essential for more comprehensive analysis. “

Comment 49:

2. They present a somewhat anecdotal approach to accepting or rejecting analytic conclusions.

AUTHORS’ RESPONSE: We regret any instances where an anecdotal approach may have been inadvertently adopted in our manuscript. In response, we have thoroughly revised the conclusion to incorporate the valuable insights raised by the reviewers. This revision includes a more analytical perspective, focusing on broader and future possibilities, as exemplified by the changes made in response to **Comments 32, 34, and 35.**

Comment 50:

3. They generate interesting new hypotheses (a connection between isolates in Okinawa, Japan and certain isolates from South America), seek evidence to confirm this possibility- and arguably make a lot out of little evidence. I believe these stand as interesting hypotheses to explore with future data.

AUTHORS' RESPONSE: We thank the reviewer for the positive remarks. We concur that the hypothesis presented warrants further exploration using future data. In accordance with this suggestion, we have been addressed in **Comment 11**.

AUTHORS' RESPONSE to Comment 11: We deeply appreciate the reviewer's suggestion regarding the inclusion of an evolutionary perspective in our study. Recognizing its importance for understanding the distribution and genetic diversity of this pathogen, we have now incorporated hypotheses related to evolutionary aspects into our discussion section. Furthermore, to gain insights into the evolutionary history, we have added an analysis of the maternally inherited apicoplast genome. This analysis, based on sequences referenced in ToxoDB61_RH, involved creating a TCS network. We used only SNPs mapped by three or more reads for this analysis, excluding five strains (FOU, GUY-KOE, MAS, P89, RUB) due to more than 5% missing data. The findings, particularly the relationship between Africa 4 (#20) and HG13, have yielded intriguing insights into the regional exchange of *T. gondii*. In addition, we have included a graphical summary illustrating the journey of Clade D *Toxoplasma gondii*, considering recombination.

Accordingly, we have changed this text to:

Apicoplast Method

(p. 28-29, line 722-725):

“The apicoplast genome referenced ToxoDB-61-*T. gondii*RH88, and a TCS network was created. For the analysis, only SNPs mapped by three or more reads were used, and five strains (FOU, GUY-KOE, MAS, P89, RUB) showing more than 5% missing sites were excluded from the analysis.”

Apicoplast Result

(p. 9, line 190-200):

“Africa 4 (#20) cluster was located at the boundary of clade D. HG13 appears genetic recombination of HG2 and Africa 4 (#20).

Additionally, an analysis of the apicoplast genome (an organelle inherited maternally) identified 177 SNPs in the 35 Kbp genome. Most Japanese strains inherited the apicoplast from HG2 (**Fig. 1B**). HG13 haplotype was linked to TgCatJp5 and HG12 (RAY and B41) by a few SNPs, suggesting their kinships. ARI shared its genome with COUG (HG11) strain. The genetic distance between HG13 and HG2 suggests that they diverged from a common ancestor and then followed different evolutionary pathways. Africa 4 (#20) lineage was distinct from other haplotypes, suggesting unique genetic characteristics.”

Evolutionary hypothesis

(p. 21-23, line 533-568):

“The type II, estimated to have appeared about 26,000 years ago¹⁶, was selected for by the domestication of the Libyan wildcat in the Fertile Crescent with the advent of agriculture about 10,000 years ago, and subsequently spread to Europe and Asia⁵⁵. It is known that agriculture also began in China and parts of Africa, suggesting that the ancestral lineages of Africa 4 (#20) and HG13 predated the arrival of type II in their respective regions^{16,56,57}. The genetic intermixing of *T. gondii* across nations and continents is believed to be due to human movement and trade^{10,58}. Trade between Eastern and Western Asia has ancient roots, with East and West Asia engaging in active trade through the Silk Road by around the 2nd BCE⁵⁹. Furthermore, from the 1st century AD onwards, maritime routes along the coasts were established, connecting China with diverse ports in Southeast Asia, India, Sri Lanka, the Arabian Peninsula, and the Horn of Africa. These maritime routes are suggested to have been the pathways through which HG13 and Africa 4 (#20) spread to coastal countries⁵⁷.

The present study does not address the topic of "how each ancestral lineage born". However, by inferring migration and admixture through the recombination history of the type II, HG13, and Africa 4 (#20) ancestral clades, we try to develop a hypothesis about the long-term migration history of clade D *T. gondii*.

- ① Initially, the HG2 lineage migrated from the Middle East via the Silk Road in the 2nd century BC, where it met and interbred with the ancestor of HG13 (**Fig. 7, red arrow**).
- ② This was followed by interbreeding with Africa 4, which migrated via the Maritime Silk Road. The exact timing of the interaction between the two lineages is unknown,

and it is possible that the order of transmission was the opposite (**Fig.7, blue arrow**). Repeated backcrossing between the hybrids and the parental species, resulting in the introgression of the HG2 and Africa 4 genome into the monochromatic ancestral population of HG13. ③ The related lineage migrated to eastern Siberia and then to North America via the Bering Strait. ④ This lineage is thought to have partially incorporated the HG11 genome, consequently evolving into HG12. The fact that large haploblocks of the Africa 4 (#20) have not mixed with lineages other than the current HG13 and some Japanese strains may be attributed to the later-arising HG2-HG13-Africa 4 admixed strain (yellow, red, blue), which swept the original HG13 lineage in East Asia. There is a possibility of discovering an ancient type of HG13 by examining *T. gondii* from wild cats in China that have not yet been isolated. ⑤ Archaeological investigations have discovered fossil of domestic cats at ancient Japanese heritage (Karakami site in Iki island, **Fig. S1**), with accelerator mass spectrometry carbon-14 dating indicating these cats date back to the 2nd century BCE⁶⁰. This is considered to mark the introduction of the domestic cat into Japan, suggesting that the *T. gondii* population arriving with these cats was admixed HG2.

(p. 44, line 1136-1141):

Figure legend:

Figure 7. Journey of Clade D *Toxoplasma* Considering Recombination.

A hypothetical diagram illustrating the historical migration events of Clade D *Toxoplasma*. The sequence of events from 1 to 5 corresponds to specific genetic recombination episodes. The red arrows represent the directions of genetic admixture. Asterisks on HG13 and HG12 indicate that these genetic lineages had ancestral-type genomes, which differed from their current genome."

Comment 51:

4. They deploy an over broad concept of Latin America. While it is true that Tropical South America (Amazonian Brazil, French Guiana) harbor populations of Toxoplasma notably lacking in clonal strains, other parts of Latin America (temperate portions of Brazil, say, and portions of the Caribbean) do have clonal

lineages and important constituents.

AUTHORS' RESPONSE: We thank for the reviewer's comment on this point. To clarify, we have added text to the Introduction as in detail in response to **Comment 16 and 37.**

AUTHORS' RESPONSE to Comment 16: We thank the reviewer for this comment. The reviewer's comment is correct. To clarify, we have added the following text to the Introduction.

(p. 5, line 90-93):

“The genetic diversity of *T. gondii* in South/Central America is quite high and differs markedly from Europe and North America(20). Although common clonal lineages (Africa I in Brazil, and type II in Chile) exist, highly genetically diverse recombinant strains are expanding⁸.”

AUTHORS' RESPONSE to Comment 37: We thank the reviewer for this comment. Accordingly, we have changed "Latin America" to "South/Central America" throughout the manuscript, including in the title, to ensure accuracy and clarity in our geographical references.

Comment 52:

60, 88, 416- it might be accurate to say that no clonal structure exists in “portions of Latin America” or, more precisely, in tropical Amazonia. But this is not true for Latin America as a whole, where temperate regions (even in Brazil) do harbor clonal strains.

AUTHORS' RESPONSE: We appreciate the reviewer's comment. We revised the manuscript as in detail in response to **Comment 51 above.**

Comment 53:

97 “It has been shown.....are thought to be” rephrase: Recently derived recombinant strains contribute.....

AUTHORS’ RESPONSE: We thank for the reviewer’s comment. We corrected the sentence as following:

(p. 5, line 101-103):

Recently derived recombinant strains significantly contribute to the current population of isolated *T. gondii*, aside from those strains that originate from distant locales and wild felines¹⁵.

Comment 54:

104 defend “a small number of ancestral lineages” or omit

AUTHORS’ RESPONSE: We thank for the reviewer’s comment. In accordance with the reviewer's comment, we have omitted this sentence from the manuscript.

Comment 55:

114 “is known to be HG1 is known” rephrase

AUTHORS’ RESPONSE: We corrected the sentence.

(p. 6, line 116):

“In Korea, the KI-1 is known to be HG1¹⁸.”

Comment 56:

123 , based on the alleles found..... sequences, (add commas)

AUTHORS' RESPONSE: We corrected the sentence.

(p. 6, line 124-127):

“A recent study investigating the genotypes of 15 isolates from Okinawa based on the alleles found for three protein-coding genes plus three intron sequences, suggested that the seven strains clustered closely with HG2, while one strain was closely related to HG3²⁴.”

Comment 57:

126 , isolated from a US AIDS patient. (add comma)

AUTHORS' RESPONSE: We corrected the sentence.

(p. 6, line 127-129):

“The remaining seven isolates formed their own cluster flanked by CAST (HG7), isolated from a US AIDS patient.”

Comment 58:

142 including isolates from China

AUTHORS' RESPONSE: In accordance with the reviewer's comment, we have changed this to:

(p. 7, line 144-147):

“In this study, we utilized POPSICLE to investigate genome-wide SNP diversity in Japanese *T. gondii* in the context of all major *T. gondii* HGs plus Africa 4(#20), including isolates from China.”

Comment 59:

168 What do you mean by “the most abundant Japanese strains?” Do you have prevalence data? Or do you just mean 3 of haplotypes?

AUTHORS’ RESPONSE: We thank for your comment. “The most abundant Japanese strains” was intended to refer the most common genotype among the Japanese strains collected in this study. Recognizing the need for clarification, we have changed this to:

(p. 8, line 177-182):

“Among the Japanese strains collected in this study, the most common genotype was clustered between HG2 and HG12 (3 out of 8: TgCatJpOk3, TgCatJpTy1/k3, and TgCatJpObi1), followed by two strains (TgBoJp1 and TgBoJp2) that were located close to the edges of HG2, and were highly similar to the reference type II ME49 genome (**Fig. 1A and Fig. S3A and B**).”

Comment 60:

250 readers do not benefit from your assessment that these programs are sophisticated and widely used, but they could benefit from a succinct description of what they do/how they work (as they did for Popsicle on line 268).

AUTHORS' RESPONSE: We wish to thank the reviewer for this comment. We agree that this point requires clarification, and have added the following text to:

(p. 12, line 272-279):

"These tools, based on the model wherein individuals are genetically derived from multiple ancestral populations, illustrate the 'mixture' of genetic ancestors within a population. The genetic composition of each individual is represented by the proportion of genetic contributions, represented as different colors, with each color corresponding to a specific ancestor. "

Comment 61:

264 This conclusion concerns me. Who decides which results deserve to be considered "correct" or "reasonable." The authors depict very flat curves for these tools, indicating that the data contain too little signal, or too much conflict, to confidently decide how many ancestral populations may have given rise to the extant data. They also show that the two tools disagree on this point. The truth may lie somewhere between, or near, 6-8.

Is this a problem with the tools? Or a reflection of uncertainty in the data? I urge the authors to consider that these models, though powerful, cannot perfectly mimic the underlying population and genetic processes.

AUTHORS' RESPONSE: We appreciate the reviewer's insightful comment on this important aspect of genetic analysis. Indeed, determining 'correct' or 'reasonable' results in such analyses poses a significant challenge. In our study, we used three statistical methods to search for the optimal number of clusters. However, the resulting flat curves from these methods indicate the difficulty in confidently identifying the exact number of ancestral populations.

Therefore, we chose to present data for different numbers of ancestors (k=6,8,14). Focusing on the changes observed in the population models and the reproducibility of known population structures from prior research, thus evaluating the effectiveness of the analytical tools.

It is important to note that the differing results yielded by each tool may not necessarily signify a flaw within the tools themselves. Instead, indicating that the data contain too little signal, or too much conflict. While these models are powerful in analyzing genetic data, they cannot perfectly mimic the complex dynamics of population and genetic processes. This situation highlights the importance of using multiple methods to cross-validate findings in population genetics.

Accordingly, we have changed this text to:

(p. 12, line 291-293):

“These results imply a limitation in the ability of these tools to accurately decipher mixed ancestry as established in earlier research^{4,15}. “

(p. 13, line 309-312):

“These three in silico methods are powerful tools, but each cannot perfectly mimic the underlying population and genetic processes. This underscores the importance of using multiple methods to cross-validate findings in population genetics.”

Comment 62:

279 Perhaps say that these results better conform to prior assumptions. Whether they “demonstrate more validity” seems to presume that they already know the answer...which seems dubious.

Also, if they embrace 14 subdivisions, I think it would be interesting to go back to fastStructure and ADMIXTURE and run those models under that assumption. Then, do these tools converge on a shared pattern of subdivision among the isolates? Figure S6 depicts using K=14 for ADMIXTURE and fastSTRUCTURE. These results deserve discussion, given the preference of the authors for that assumption.

AUTHORS’ RESPONSE: We wish to thank the reviewer for this comment. The results of “K=14” in ADMIXTURE and fastSTRUCTURE segmented each HGs, but

did not seem to resolve the admixture of genetic ancestry within the population. In accordance with the reviewer's comment, we have revised this text to:

(p. 12, line 306-309):

“In contrast, the results of “K=14” in ADMIXTURE and fastSTRUCTURE segmented each HG, but did not seem to resolve the admixture of genetic ancestry within the population (**Fig. S8B and D**). Taken together, these results suggest that POPSICLE more closely conform to prior assumptions. “

Comment 63:

282 This logic confuses me. Explain how including other, less related populations, would undermine the goal? By placing all of the Japanese isolates into one or just a few groups?

287 Again I wonder about the criteria....who decides when subdivision is “unnecessary” and on what grounds? Aesthetic preference? Going back to the previous comment, if you added less closely related lineages, does the tool now group these three HG15 lineages?

AUTHORS’ RESPONSE: We greatly appreciate the reviewer's comment on this point. We have clarified this oversight phrase and have revised the manuscript.

The feature of POPSICLE is to visualize genomic structures by assigning unique colors to each ancestral lineage. While it is technically feasible to include all 57 strains in the POPSICLE analysis, as the number of ancestral lineages increases, the colors become more subdivided, leading to decreased visibility. To mitigate this issue and produce clearer plot, we strategically excluded ancestral lineages with low relevance to Japanese strains from our analysis. The selection was based on the results of POPSICLE, created using various patterns, including data substitution and resampling (**Fig. S10A-C**). We then used the minimal set of ancestral populations necessary to depict the genomic

structure of the *T. gondii* population in Far East Asia. We confirmed that even with the addition of distant lineages, the primary genomic structure of the Far East Asian population as depicted was mostly conserved.

In the plot of Fig. 5D, when other lineages (HG5 and HG10) were added, two strains of HG15 (TgRsCr01 and TgCtCo5) were identified as a mixed strain of HG6 (deep green), HG5 and HG10 (khaki), and TgCatJp5 (cyan), resulting in the grouping of HG15 into one category.

We have added text to:

(p. 13, line 313-321):

“The feature of POPSICLE is to visualize genomic structures by assigning unique colors to each ancestral lineage. While it is technically feasible to include all 57 strains in the POPSICLE analysis, as the number of ancestral lineages increases, the colors become more subdivided, leading to decreased visibility. To mitigate this issue and produce clearer plot, we strategically excluded ancestral lineages with low relevance to Japanese strains from our analysis. The dataset selected was based on the results of POPSICLE, created using various patterns, including data substitution and resampling. Its robustness was validated. We then used the minimal set of ancestral populations necessary to depict the genomic structure of the *Toxoplasma* population in Far East Asia.”

Comment 64:

293 HG3, a recombinant line,

AUTHORS’ RESPONSE: We express our gratitude for the reviewer’s comment. To ensure we have correctly understood the point made, are you suggesting that admixed strains such as B73 and TgDgCo17 should be distinctly separated from HG1 and HG3? In accordance with the reviewer's comment, we have changed this text to:

(p. 14, line 328-330):

To verify that the POPSICLE software was performing as expected, we analyzed genome-data derived from archetypal I, II, and III and 2 recombinant lines (B73 and TgDgCo17), where admixture among ancestors is well-documented (**Fig. 3B**)^{15,39}.

Comment 65:

377 Do you take this as the basis of a conclusion, or the basis of establishing a hypothesis? I enjoyed this “part of the journey” but I am a little uncomfortable about the seemingly post hoc nature of seeking confirmation for a particular connection here.

AUTHORS’ RESPONSE: We appreciate the reviewer’s comment on this point. We took this as the basis for establishing a hypothesis. Given that our data set comprises only a limited number of samples from Okinawa and lacks samples from other Southeast Asian regions, we emphasize that the hypothesis suggesting a shared Jp5^{cyan} ancestor remains to be validated with more comprehensive sampling.

In accordance with the reviewer's comment, we have changed this text to:

(p. 17, line 418-422):

“The HG5 and HG10 genomes were resolved as a mosaic pattern with a major ancestry (khaki) alongside many other colors. Although the locations were on different chromosomal coordinates (5-32 blocks of 10 kbp window size), all strains of HG5 and HG10 had small blocks of Jp5^{cyan} (**Fig. 5E and Fig. S13**), suggesting potential shared ancestry (**Fig. 5F**).”

In addition, we are grateful for the insightful scientific comment provided. We acknowledge the importance of a detailed analysis of genetic diversity within lineages to better understand the history of divergence within the same group. In response, we

have conducted an additional Neighbor-net analysis, that includes type II, type X, and HG2-like Japanese strains (see **Fig. S3C**).

Accordingly, we have added:

(p. 8-9, line 182-185):

"An intra-group analysis of type II, HG12, and HG2-like Japanese strains showed that three strains—TgCatJpOk3, TgCatJpTy1/k3, and TgCatJpObi1—were distinct from both type II and HG12 strains (**Fig. S3C**). Among them, TgCatJpObi1 specifically exhibited evidence of recombination with HG12."

(p. 23, line 589-593):

"Furthermore, the dataset employed in the present study was insufficient for the fine-scale genetic distances or specific timing of divergence between intra-lineage strains. Future analysis with a larger number of isolates within the same group will provide a better understanding of the genetic microstructure. This study lays the groundwork for such comprehensive future research."

Moreover, we recognize several biases should be considered as limitations of this study. One primary challenge is the representativeness of our sampling. Currently, genomic data from West Asia, Central Asia, Southeast Asia, and Russia especially, the Far-East Asian region are absent, leaving a significant gap in covering the genetic diversity present across the vast Asian continent. This gap limits our ability to determine the continuity of certain haplogroups (HG2-HG13, HG13-HG12) across different regions. Therefore, collecting samples from these areas is essential for more comprehensive analysis. Additionally, a better understanding the distribution of genotypes such as TgCatJpOk4^{Magenta} and TgCatJp5^{Cyan}, which so far have been identified only in Japan within Asia, is crucial to grasp the full scope of *T. gondii*'s spread and propagation. This study represents an initial step in integrating the *T. gondii* genome from East Asia into global knowledge and how future research can address these gaps.

Accordingly, we have changed this text to:

(p. 23, line 582- 588):

“We recognize several biases should be considered as limitations of this study. One primary challenge is the representativeness of our sampling. Currently, genomic data from West Asia, Central Asia, Southeast Asia, and Russia are absent, leaving a significant gap in covering the genetic diversity present across the vast Asian continent. This gap limits our ability to determine the continuity of certain haplogroups (HG2-HG13, HG13-HG12) across different regions. Therefore, collecting samples from these areas is essential for more comprehensive analysis. “

Comment 66:

383 fairer to say that Asia was underrepresented. I would not agree that all regions except Asia have enjoyed careful attention. Much of Africa, for example, remains poorly characterized.

AUTHORS’ RESPONSE: We thank the reviewer for this comment. Accordingly, we have changed this to:

(p. 19, line 446-450):

“Previous work was performed to describe the population genetic structure of *T. gondii* at whole genome resolution. However, since number of *T. gondii* strains from Africa and Asia are insufficient, their high-resolution genetic structures remain unclear. In particular, Asia was underrepresented as only one isolate, TgCatPRC2, has been included in all previous large-scale population genomic studies^{4,15,16}.”

Comment 68:

385-8 The addition of these several isolates from Japan and China are very welcome. But I think, in fairness, the authors should acknowledge this is a good start. But Asia is vast, and a mere handful of isolates will likely bias inferences of population diversity and structure.

AUTHORS' RESPONSE: We thank for the reviewer's comment. We acknowledged that this study handles a small number of isolates in Japan and China within a vast Asia. In the limitations section, we mentioned that future data will reveal a better population structure. The changes are written in **Comment 65 above**.

Comment 69:

421 while plausible, this idea really needs tempering by acknowledging the very small sample here. Okinawa may have maintained an isolated lineage. But where have we still not sampled? Will our view change?

AUTHORS' RESPONSE: We strongly appreciate the reviewer's comment on this point. Accordingly, we have added text:

(p. 23-24, line 593-602):

“Additionally, the samples from Western Japan used in this study were limited to only two strains isolated in in the Shikoku region, excluding Okinawa. Therefore, the Jp5^{cyan} lineage may also exist in the Kyusyu region, which is geographically close to Okinawa. Furthermore, the distance between the Okinawa Islands and Taiwan or Shanghai is similar to that of the Kyusyu region in Japan, suggesting the possibility that this lineage could have been introduced via Southeast Asia or China. It would be helpful to study how common the Jp5^{cyan} lineage of *T. gondii* is in Southeast Asia. This will aid in understanding the spread of the parasite. This study is a first step in adding information about the *T. gondii* genome from East Asia to the global knowledge base.”

Comment 70:

One technique the authors have not used is to resample data, with replacement. How stable is their view to stochastic variation in the present dataset? Would removing some data or some isolates reshape the view, or are the highlighted findings robust to

such perturbations? The Haplotype network (554) was constructed with bootstrapping as was the RAxML analysis of ROP5 and ROP18 (597). What about other results.

AUTHORS' RESPONSE: We thank you for the reviewer's insightful comment. Random numbers were generated, after excluding four strains each that were selected, for new POPSICLE plots were created. Even excluding these randomly selected isolates, the interpretation of inter-lineage admixtures was stable (**Fig. 10B and C**).

Comment 71:

436 Yes I like this approach.... Here you illustrate what questions future data can help resolve. I think this paper would benefit by adopting this perspective more consistently.

AUTHORS' RESPONSE: We thank the reviewer for the positive comment. Our research represents the first initiative to integrate the genome insight of *Toxoplasma* from East Asia into global structure, proposing a hypothesis on the biographical nexus of Africa, Asia, and American *Toxoplasma*. Although this study is based on a limited number of strains from Japan and China, it plays an important role in illustrating how future data can contribute in resolving outstanding questions. We have revised our discussion with this in mind. **e.g. Comment 69 above.**

Comment 72:

454 remove "the" in front of 165 cases

AUTHORS' RESPONSE: We corrected this point.

(p. 21, line 521-523):

“However, out of the 166 cases of symptomatic porcine toxoplasmosis diagnosed during slaughter inspections in Japan over the past five years (2018-2022), 165 cases were identified in Okinawa.”

REVIEWERS' COMMENTS

Reviewer #1 (Remarks to the Author):

The authors have carefully addressed all my comments.

I have a last comment regarding the diagram proposing a scenario on the historical migration events of Clade D. Land migrations across the Bering Strait were feasible solely in glacial epochs when substantial ice sheets established a land bridge linking the two continents. Does this scenario consider this point?

Reviewer #2 (Remarks to the Author):

In this manuscript, Ihara et al describe the findings of their analysis of the population structure of *Toxoplasma gondii* in Japan by applying whole genome sequencing on 15 parasite strains isolated from Japan, seven of them using a novel target-enrichment approach, plus two strains from China. Using a combination of several known bioinformatic tools commonly used for population structure studies, including a relatively new program called POPSICLE, the authors show evidence of the presence of two novel ancestral Japanese parasite lineages, one sharing a common ancestor with haplogroup (HG) 11 and the second lineage, apparently confined to the Japanese island of Okinawa, also found in strains from Central/South America. High-resolution genome-wide *T. gondii* population structure studies from Far-East Asia has been lacking and the work from Ihara et al fills this gap in knowledge and provides new insights into the dissemination and evolution of this important parasite. The results and findings presented herein will be of interest for the research communities working on public health issues associated with Toxoplasmosis and other related apicomplexan parasites, as well as in the fields of parasite evolution and ecology. I find the authors have addressed all my previous concerns and therefore, I recommend the manuscript for its publication in Nature Communications.

Minor edits:

- 1- Fig. 1A, Fig S3A and Fig S3C: TgCatJpObi1 name seems to be truncated.
- 2- Fig 2C and Supp Fig 7C, please indicate what the dashed lines represent in the legend.
- 3- Names in Neighbor-network trees are too small and difficult to read.

Reviewer #2 (Remarks on code availability):

The code contains enough comments and accompanying documentation to run the analyses described in the main manuscript.

Reviewer #3 (Remarks to the Author):

I believe the authors have faithfully and competently addressed all concerns raised in review and complement them for doing so.

Point-by-point responses to Reviewer #1:

Reviewer #1 (Remarks to the Author):

The authors have carefully addressed all my comments.

I have a last comment regarding the diagram proposing a scenario on the historical migration events of Clade D. Land migrations across the Bering Strait were feasible solely in glacial epochs when substantial ice sheets established a land bridge linking the two continents. Does this scenario consider this point?

AUTHORS' RESPONSE: We thank the reviewer for the insightful comment on the historical migration events of *T. gondii*. We appreciate your attention to the detail of the proposed scenario, specifically the feasibility of land migrations across the Bering Strait during glacial epochs. We would like to add below text regarding the movement across the Bering strait.

Accordingly, we have changed related text to:

(p. 23, line 555-line 567):

“**③**The related lineage migrated to eastern Siberia and then to North America via the Bering Strait. Land animals were only able to migrate across the Bering Strait during the glacial epochs, when substantial ice sheets established a land bridge between the two continents. However, the last land bridge formed more than 10,000 years ago, and the timing of the land bridge's existence may differ from the timing of the emergence of the ancestral lineage of HG12 (a HG2-HG13 admixture). This suggests that the migration of ancestral HG12 to North America might not have occurred via the Bering land bridge, and that humans, migratory birds, or marine mammals may have been responsible for this migration. While HG12 is an admixed strain of HG2 and HG13, it shows almost no evidence of admixture with the Africa 4 lineage. It should be noted that this does not indicate that the divergence from HG2-HG13 to HG12 occurred specifically between points **①** and **②**; it only broadly suggests the process of HG12's formation. Future isolation and genomic analysis of *T. gondii* on either the Eurasian or Alaskan side near the Bering Strait may further clarify the pathway.”

Point-by-point responses to Reviewer #2:

Reviewer #2 (Remarks to the Author):

*In this manuscript, Ihara et al describe the findings of their analysis of the population structure of *Toxoplasma gondii* in Japan by applying whole genome sequencing on 15 parasite strains isolated from Japan, seven of them using a novel target-enrichment approach, plus two strains from China. Using a combination of several known bioinformatic tools commonly used for population structure studies, including a relatively new program called POPSICLE, the authors show evidence of the presence of two novel ancestral Japanese parasite lineages, one sharing a common ancestor with haplogroup (HG) 11 and the second lineage, apparently confined to the Japanese island of Okinawa, also found in strains from Central/South America. High-resolution genome-wide *T. gondii* population structure studies from Far-East Asia has been lacking and the work from Ihara et al fills this gap in knowledge and provides new insights into the dissemination and evolution of this important parasite. The results and findings presented herein will be of interest for the research communities working on public health issues associated with Toxoplasmosis and other related apicomplexan parasites, as well as in the fields of parasite evolution and ecology. I find the authors have addressed all my previous concerns and therefore, I recommend the manuscript for its publication in Nature Communications.*

AUTHORS' RESPONSE: We thank the reviewer for these excellent comments.

Minor edits:

1- Fig. 1A, Fig S3A and Fig S3C: TgCatJpObi1 name seems to be truncated.

AUTHORS' RESPONSE: We appreciate the careful review of the manuscript. We have corrected those labels in Fig.1a, Fig S3b and Fig S3c.

2- Fig 2C and Supp Fig 7C, please indicate what the dashed lines represent in the legend.

AUTHORS' RESPONSE: We thank the reviewer for the careful review. We have added a sentence in legends for Fig 2c and Fig S7c.

(p. 54, line 1251-1253):

“Dashed lines represent estimated copy depth of the archetypes strains (blue, type I/RH-88; green, type II/ME49; red, type III/VEG).”

3- Names in Neighbor-network trees are too small and difficult to read.

AUTHORS’ RESPONSE: We appreciate these helpful suggestions. We have enlarged labels on Neighbor-network tree for Fig. 1, 4, 5, and 6.